# Conformalized Credal Set Predictors

**Alireza Javanmardi**
LMU Munich, MCML
Munich, Germany
alireza.javanmardi@ifi.lmu.de

**David Stutz**
Max Planck Institute for Informatics
Saarbrücken, Germany
david.stutz@mpi-inf.mpg.de

**Eyke Hüllermeier**
LMU Munich, MCML
Munich, Germany
eyke@ifi.lmu.de

## Abstract

Credal sets are sets of probability distributions that are considered as candidates for an imprecisely known ground-truth distribution. In machine learning, they have recently attracted attention as an appealing formalism for uncertainty representation, in particular, due to their ability to represent both the aleatoric and epistemic uncertainty in a prediction. However, the design of methods for learning credal set predictors remains a challenging problem. In this paper, we make use of conformal prediction for this purpose. More specifically, we propose a method for predicting credal sets in the classification task, given training data labeled by probability distributions. Since our method inherits the coverage guarantees of conformal prediction, our conformal credal sets are guaranteed to be valid with high probability (without any assumptions on model or distribution). We demonstrate the applicability of our method on ambiguous classification tasks for uncertainty quantification.

## 1 Introduction

Representing and quantifying uncertainty is becoming increasingly important in machine learning (ML), particularly as ML models are employed in safety-critical application domains such as medicine or autonomous driving. In such domains, a distinction between so-called *aleatoric uncertainty* (AU) and *epistemic uncertainty* (EU) is often useful [21]. Broadly speaking, aleatoric uncertainty is due to the inherent randomness of the data-generating process, whereas epistemic uncertainty stems from the learner's lack of knowledge about the best predictive model. Thus, while the former is irreducible, the latter can, in principle, be reduced through additional information, e.g., by gathering additional data to learn from.

Representation of aleatoric and epistemic uncertainty requires formalism more expressive than standard probability distributions [22]. One such formalism which prevails in the recent ML literature is second-order probability distributions. Essentially, in a classification setting, these are distributions over distributions over classes. Models producing second-order distributions as predictions can be learned in a classical Bayesian way [16, 25] or using more recent approaches such as evidential deep learning [44]. Yet, approaches of that kind are not unproblematic and have been subject to criticism [8, 9]. Specifically, such approaches have been shown to misrepresent epistemic uncertainty. Another formalism suitable for representing both types of uncertainty is the concept of a *credal set*, which is well-established in the field of imprecise probability theory [58] and meanwhile also attracted attention in ML [23, 46]. Credal sets are (convex) sets of probability distributions that can be considered as candidates for an imprecisely known ground-truth distribution.

38th Conference on Neural Information Processing Systems (NeurIPS 2024).

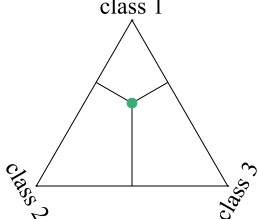 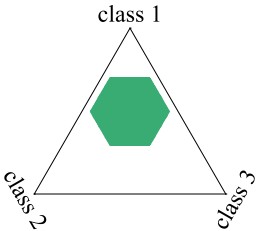 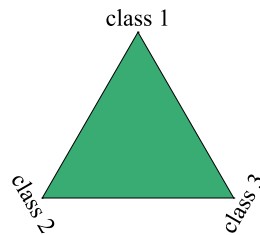

Figure 1: For the three-class classification setting, the space of probability distributions can be illustrated by a two-dimensional simplex: each point in the simplex corresponds to a probability distribution so that credal sets can be depicted as regions. The left case corresponds to the special case of a singleton (credal) set, i.e., a precise probability distribution, signifying aleatoric but no epistemic uncertainty. The case in the middle represents partial knowledge with a certain degree of (epistemic) uncertainty about the true distribution, and the right one corresponds to the case of complete ignorance, where nothing is known about the distribution.

Figure 1 shows examples of credal sets in a three-class scenario, where the space of distributions can be visualized by the two-dimensional probability simplex. Broadly speaking, the larger the credal set, the higher the epistemic uncertainty, and the more "in the middle" the set is located, i.e., the closer it is to the uniform distribution, the higher the aleatoric uncertainty.

Learning to predict second-order representations, such as credal sets or second-order distributions, from standard "zero-order" data — training instances together with observed class labels — has been shown to be difficult in that it typically provides biased estimates of epistemic uncertainty [8, 9]. To alleviate this problem, we assume "first-order" training data, i.e., instances associated with probability (frequency) distributions over the class labels. In other words, instances are labeled probabilistically instead of being assigned a deterministic class label. This type of data is becoming increasingly available in practice, for example, in the form of aggregations over multiple annotations per data instance [10, 31, 35, 63], and hence increasingly relevant in many applications [52, 53]. Moreover, such data facilitates second-order learning.

In this paper, we leverage conformal prediction (CP), a non-parametric approach for set-valued prediction rooted in classical frequentist statistics [57], to construct credal sets. With relatively mild assumptions similar to those in CP, this approach inherits the so-called marginal coverage from CP: Predicted sets are guaranteed to cover the true target with high probability. For us, this means that we can use any first-order or second-order predictive model to construct unbiased credal sets equipped with such a coverage guarantee. Specifically, we propose various nonconformity functions applicable to first- and second-order predictors to construct conformal credal sets. We also study the case where only a noisy version of first-order distributions is available, demonstrating that the coverage guarantee holds under a bounded noise assumption. On ChaosNLI [31], an ambiguous natural language inference task with multiple annotations per example, we show that our conformal credal sets are valid, i.e., covering the true ground-truth distribution with high probability while comparing the efficiency of different nonconformity functions. Together with experiments on CIFAR10-H [35]—a variant of the CIFAR10 test set containing class distributions from human annotations—we demonstrate how these credal sets enable practical quantification of aleatoric and epistemic uncertainty. We further complement this study with controlled experiments on synthetic data, specifically investigating the performance of credal set prediction in the presence of noisy data.

## 2 Background

### 2.1 Supervised Learning and Predictive Uncertainty

We consider the setting of (polychotomous) classification with label space $\mathcal{Y} = \{1, \ldots, K\}$ and an instance space $\mathcal{X}$. As usual, we assume an underlying data-generating process in the form of a probability distribution $P$ on $\mathcal{X} \times \mathcal{Y}$, so that observations $(X, Y)$ are i.i.d. samples from $P$. We denote by $\boldsymbol{\lambda}^{\boldsymbol{x}} = (\lambda_1^{\boldsymbol{x}}, \ldots, \lambda_K^{\boldsymbol{x}})^\top \in \Delta^K$ the conditional probability distribution $P(\cdot \mid X = \boldsymbol{x})$, which we also consider as an element of the $(K-1)$-simplex $\Delta^K$. Thus, the probability to observe $Y = k$ as an outcome for $\boldsymbol{x} \in \mathcal{X}$ is given by $\lambda_k^{\boldsymbol{x}}$.

Since the dependency between instances $X$ and outcomes $Y$ is non-deterministic, the prediction of $Y$ given $X = \boldsymbol{x}$ is necessarily afflicted with uncertainty, even if the ground-truth distribution $\boldsymbol{\lambda^x}$ is known. As already said, this uncertainty is commonly referred to as *aleatoric* [22]. Intuitively, the closer $\boldsymbol{\lambda^x}$ to the uniform distribution $\boldsymbol{p}_{\text{uni}} = (1/K, \dots, 1/K)^\top$, the higher the uncertainty, and the closer it is to a degenerate (Dirac) distribution assigning all probability mass to a single class (a corner point in $\Delta^K$), the lower the uncertainty.

Instead of assuming $\boldsymbol{\lambda^x}$ to be known, suppose now that only a prediction $\hat{\boldsymbol{\lambda}}^{\boldsymbol{x}}$ of this distribution is available. *Epistemic uncertainty* refers to the uncertainty about how well the latter approximates the former, and hence to the additional uncertainty in the prediction of outcome $Y$ that is caused by the discrepancy between $\hat{\boldsymbol{\lambda}}^{\boldsymbol{x}}$ and the ground-truth $\boldsymbol{\lambda^x}$. We seek to capture this discrepancy by means of credal sets $Q \in \mathcal{Q}_K \subset \Delta^K$, with the idea that $Q \ni \boldsymbol{\lambda^x}$ holds with high probability. Typically, credal sets are assumed to be convex, and further restrictions might be imposed on $\mathcal{Q}_K$ for practical and computational reasons, for example, a restriction to convex polygons (with a finite number of extreme points).

## 2.2 Conformal Prediction

Conformal prediction provides a general framework for producing set-valued predictions with a certain guarantee of validity. In a supervised setting, consider data points of the form $Z = (X, U) \in \mathcal{X} \times \mathcal{U}$, and the task is to predict $U$ given $X = \boldsymbol{x}$. We assume the space $\mathcal{Z} = \mathcal{X} \times \mathcal{U}$ to be equipped with a nonconformity measure $f : \mathcal{Z} \longrightarrow \mathbb{R}$ that quantifies the "strangeness" of $\boldsymbol{z}$, i.e., the higher $f(\boldsymbol{z})$, the less normal or expected the data point. Let $\mathcal{D}_{\text{calib}} \subset \mathcal{Z}$ be a (randomly generated) set of data points, called *calibration data*, and $Z$ another data point that remains unobserved. Under the assumption of exchangeability, i.e., that the calibration data and the query point $Z$ have been generated by an exchangeable process, we seek a so-called confidence set $C \subseteq \mathcal{U}$ that guarantees coverage:

$$\mathbb{P}(U \in C) \geq 1 - \alpha. \tag{1}$$

By a simple combinatorial argument [57], the confidence set $C$ can be constructed as

$$C(\boldsymbol{x}) := \left\{ u \in \mathcal{U} \mid f(\boldsymbol{x}, u) \leq q(\mathcal{E}, \alpha') \right\}, \tag{2}$$

where $\mathcal{E} := \{ f(\boldsymbol{z}) : \boldsymbol{z} \in \mathcal{D}_{\text{calib}} \}$ is the set of nonconformity scores, $\alpha' = |\mathcal{E}|^{-1} \lceil (1 + |\mathcal{E}|)(1 - \alpha) \rceil$, and $q(\mathcal{E}; \alpha')$ denotes the $\alpha'$-quantile of $\mathcal{E}$. Importantly, the guarantee (1) holds regardless of the nonconformity function $f(\cdot)$, which, however, has an influence on the *efficiency* of the prediction: The more appropriate the function, the smaller the prediction set $C$ tends to be. Normally, $f(\cdot)$ is not predefined but constructed in a data-driven way using training data $\mathcal{D}_{\text{train}}$. For example, a common approach is to train a predictor $\pi : \mathcal{X} \longrightarrow \mathcal{U}$ and then define $f(\boldsymbol{x}, u)$ in terms of $d(u, \pi(\boldsymbol{x}))$, where $d(\cdot, \cdot)$ is an appropriate distance function on $\mathcal{U}$. Replacing the point-prediction $\pi(\boldsymbol{x}) \in \mathcal{U}$ by the prediction set $C(\boldsymbol{x}) \subset \mathcal{U}$ can then be seen as "conformalizing" the predictor $\pi$.

# 3 Conformal Credal Set Prediction

Our goal is to learn a credal set predictor $h : \mathcal{X} \longrightarrow \mathcal{Q}_K$, that is, a model that makes predictions in the form of credal sets, thereby representing both aleatoric and epistemic uncertainty. To this end, we assume access to first-order data, i.e., probabilistic training data of the form

$$\mathcal{D} = \left\{ \left( \boldsymbol{x}_1, \boldsymbol{\lambda}^{\boldsymbol{x}_1} \right), \dots, \left( \boldsymbol{x}_N, \boldsymbol{\lambda}^{\boldsymbol{x}_N} \right) \right\} \subset \mathcal{X} \times \Delta^K. \tag{3}$$

The model $h$ should be able to predict the (probabilistic) outcomes for new query instances in a reliable way. More specifically, suppose that $\boldsymbol{x}_{\text{new}}$ is a new query instance (following the same distribution as the training data) for which a prediction is sought. To correctly represent epistemic and aleatoric uncertainty, we want the credal prediction $Q = h(\boldsymbol{x}_{\text{new}})$ to be valid, meaning $Q \ni \boldsymbol{\lambda}^{\boldsymbol{x}_{\text{new}}}$ with high probability, while at the same time being informative such that the (epistemic) uncertainty reflected by $Q$ is as small as possible.

We aim to construct the credal set predictor $h$ by means of conformal prediction. Following the conformalization recipe outlined in Section 2.2, we partition $\mathcal{D}$ into $\mathcal{D}_{\text{train}}$ and $\mathcal{D}_{\text{calib}}$, using the former for model training and the latter for calibration.

---

**Algorithm 1** Conformal Credal Set Prediction

---

**Input:**

       Data $\mathcal{D}$; error rate $\alpha$; query instance $\boldsymbol{x}_{\text{new}}$.

**Process:**

       Partition $\mathcal{D}$ into $\mathcal{D}_{\text{train}}$ and $\mathcal{D}_{\text{calib}}$.
       Train a first-order ($i = 1$) or a second-order ($i = 2$) predictor using $\mathcal{D}_{\text{train}}$.
       Choose a nonconformity function $f_i$ as in (5) or (8) that suits the trained predictor to obtain the set of scores $\mathcal{E}_i$.
       Set $\alpha' = |\mathcal{E}_i|^{-1}\lceil(1 + |\mathcal{E}_i|)(1 - \alpha)\rceil$.

**Output:**

       $h_i(\boldsymbol{x}_{\text{new}}) = \big\{ \boldsymbol{\lambda} \in \Delta^K \mid f_i(\boldsymbol{x}_{\text{new}}, \boldsymbol{\lambda}) \le q(\mathcal{E}_i, \alpha') \big\}.$

---

Regarding the training step, we explore two learning strategies connected with two ways of defining a nonconformity function, which is pivotal in the calibration step. The first approach is based on training a standard (*first-order*) probability predictor, i.e., a probabilistic classifier $g : \mathcal{X} \longrightarrow \Delta^K$ that maps instances to the (first-order) probability distribution on $\mathcal{Y}$. This can be achieved, for example, by minimizing the cross-entropy loss between the ground truth and the predicted distributions, i.e.,

$$g = \underset{\bar{g} \in \mathcal{H}}{\operatorname{argmin}} \sum_{(\boldsymbol{x}_i, \boldsymbol{\lambda}^{\boldsymbol{x}_i}) \in \mathcal{D}_{\text{train}}} - \sum_{k=1}^{K} \lambda_k^{\boldsymbol{x}_i} \log(\bar{g}(\boldsymbol{x}_i)_k), \tag{4}$$

where $\mathcal{H}$ is a hypothesis space. Given a predictor $g(\cdot)$ of this kind, nonconformity is naturally defined in terms of a distance:

$$f_1(\boldsymbol{x}, \boldsymbol{\lambda}^{\boldsymbol{x}}) := d(\boldsymbol{\lambda}^{\boldsymbol{x}}, g(\boldsymbol{x})), \tag{5}$$

where $d(\cdot, \cdot)$ is a suitable distance function on $\Delta^K$, such as total variation, Wasserstein distance, etc.

An alternative approach is motivated by recent work on (epistemic) uncertainty representation via *second-order* probability distributions [16, 25, 44]. A second-order learner $G : \mathcal{X} \longrightarrow \mathbb{P}(\Delta^K)$ maps each input $\boldsymbol{x}$ to a distribution over $\Delta^K$. Given the training data of the form (3), meaningful learning in this context can be accomplished, for instance, by parameterizing the second-order distributions using Dirichlet distributions. Specifically, one can assume that each $\boldsymbol{x}$ is associated with a Dirichlet distribution characterized by the parameter vector $\boldsymbol{\theta}^{\boldsymbol{x}} \in \mathbb{R}_{\ge 1}^K$ with the probability density function

$$P(\boldsymbol{\lambda} \mid \boldsymbol{\theta}^{\boldsymbol{x}}) = \frac{1}{B(\boldsymbol{\theta}^{\boldsymbol{x}})} \prod_{k=1}^{K} \lambda_k^{\theta_k^{\boldsymbol{x}} - 1}, \tag{6}$$

where $B(\cdot)$ is the multivariate beta function. This way, $\boldsymbol{\lambda}^{\boldsymbol{x}}$ can be thought of as a sample from that distribution, i.e., $\boldsymbol{\lambda}^{\boldsymbol{x}} \sim \text{Dir}(\boldsymbol{\theta}^{\boldsymbol{x}})$. The model aims to find parameter vectors $\boldsymbol{\theta}^{\boldsymbol{x}}$s that minimize the negative log-likelihood loss

$$\sum_{(\boldsymbol{x}_i, \boldsymbol{\lambda}^{\boldsymbol{x}_i}) \in \mathcal{D}_{\text{train}}} \left( \log(B(\boldsymbol{\theta}^{\boldsymbol{x}_i})) - \sum_{k=1}^{K} (\theta_k^{\boldsymbol{x}_i} - 1) \log(\lambda_k^{\boldsymbol{x}_i}) \right). \tag{7}$$

Given a second-order predictor $\boldsymbol{\theta}^{\boldsymbol{x}}$, nonconformity can be defined as a decreasing function of likelihood, e.g., as 1 minus relative likelihood:

$$f_2(\boldsymbol{x}, \boldsymbol{\lambda}^{\boldsymbol{x}}) = 1 - \frac{P(\boldsymbol{\lambda}^{\boldsymbol{x}} \mid \boldsymbol{\theta}^{\boldsymbol{x}})}{\max_{\boldsymbol{\lambda} \in \Delta^K} P(\boldsymbol{\lambda} \mid \boldsymbol{\theta}^{\boldsymbol{x}})}. \tag{8}$$

Using the nonconformity function $f_i(\cdot)$ ($i \in \{1, 2\}$), we obtain the set of nonconformity scores by

$$\mathcal{E}_i := \Big\{ f_i(\boldsymbol{x}_j, \boldsymbol{\lambda}^{\boldsymbol{x}_j}) \mid (\boldsymbol{x}_j, \boldsymbol{\lambda}^{\boldsymbol{x}_j}) \in \mathcal{D}_{\text{calib}} \Big\}. \tag{9}$$

Accordingly, the credal set can be defined as

$$h_i(\boldsymbol{x}_{\text{new}}) := \big\{ \boldsymbol{\lambda} \in \Delta^K \mid f_i(\boldsymbol{x}_{\text{new}}, \boldsymbol{\lambda}) \le q(\mathcal{E}_i, \alpha') \big\}. \tag{10}$$

Algorithm 1 outlines a summary of the proposed methods. In the following theorem, we state the validity of the predicted set, that is, the restatement of the conformal coverage guarantee [57] adjusted to our setting.

**Theorem 3.1** (Validty Gaurantee). *Let $\mathcal{P}$ denote the joint probability distribution on $(X, \Lambda) \in \mathcal{X} \times \Delta^K$. If data points in $\mathcal{D}_{calib}$ and $(\boldsymbol{x}_{new}, \boldsymbol{\lambda}^{\boldsymbol{x}_{new}})$ are drawn exchangeably from $\mathcal{P}$, then the conformal credal sets in* (10) *are valid, i.e.,*

$$\mathbb{P}\big(\boldsymbol{\lambda}^{\boldsymbol{x}_{new}} \in h_i(\boldsymbol{x}_{new})\big) \geq 1 - \alpha\,, \ \ for \ i \in \{1, 2\}.$$

It is worth mentioning that both predictors can also be trained using zero-order data. A first-order predictor can be achieved through standard training with a cross-entropy loss function, while a second-order predictor can be achieved, for instance, through the means of evidential learning [44]. Hence, it is sufficient to have probabilistic calibration data in order to have a reasonable judgment of the predictors' performances and be able to conformalize them.

Moreover, while these two approaches can be compared in various ways, one immediate observation is that training a second-order predictor poses greater challenges. However, credal sets constructed using the second-order predictor exhibit a natural and superior adaptivity compared to those constructed by the first-order predictor. This is because, with the first-order predictor, once the quantile is determined during calibration, all distributions within a certain distance are included in the set for any given point at prediction time. On the other hand, with the second-order predictor, the resulting set depends on the calculated quantile as well as the skewness of the predicted distribution.

### 3.1 Generalization to Imprecise First-Order Data

So far, we (implicitly) assumed that ground-truth probability distributions $\boldsymbol{\lambda}^{\boldsymbol{x}_i}$ will be provided as calibration (and training) data. Needless to say, this assumption will rarely hold true in practice. Instead, observations will rather be noisy versions $\tilde{\boldsymbol{\lambda}}^{\boldsymbol{x}_i}$ of the true probabilities. Notably, such datasets emerge in scenarios where each data instance $\boldsymbol{x}$ is annotated by multiple human experts, which recently have attracted a lot of attention in the context of machine learning [10, 31, 35, 43, 63] and also conformal prediction [24, 52]. In this context, $\tilde{\boldsymbol{\lambda}}^{\boldsymbol{x}}$ denotes the distribution derived from aggregating annotator disagreements concerning the label of instance $\boldsymbol{x}$. Of course, conformal prediction can still be applied to noisy data of that kind, but the coverage guarantee will then only hold w.r.t. noisy labeling, i.e., $\mathbb{P}(\tilde{\boldsymbol{\lambda}}^{\boldsymbol{x}_{new}} \in h(\boldsymbol{x}_{new})) \geq 1 - \alpha$.

Practically, one may expect that the guarantees will hold for the ground truth as well, simply because calibration on noisy instead of clean data will tend to make prediction regions larger and hence more conservative. Moreover, since nonconformity is derived from a predictive model $g(\cdot)$ that seeks to recover ground-truth probabilities, the latter should conform at least as well as noisy distributions. Of course, this intuition is not a formal guarantee. In order to provide such a guarantee for the ground-truth probabilities, one obviously needs to make some assumptions. Concretely, let us make the following *bounded noise* assumption for the labeling process: The labeling noise is (stochastically) bounded in the sense that, given the nonconformity function $f$ and a (small) probability $\delta > 0$, there exists a tolerance $\epsilon > 0$ such that the following holds all $\boldsymbol{x} \in \mathcal{X}$:

$$\mathbb{P}\left(|f(\boldsymbol{x}, \boldsymbol{\lambda}^{\boldsymbol{x}}) - f(\boldsymbol{x}, \tilde{\boldsymbol{\lambda}}^{\boldsymbol{x}})| < \epsilon\right) \geq 1 - \delta. \tag{11}$$

**Theorem 3.2.** *Let $\alpha > 0$ be any miscoverage rate, and suppose the bounded noise assumption holds. Let $q = q(\mathcal{E}, \tilde{\alpha})$ be the critical threshold on the noisy calibration data $\mathcal{D}_{calib}$ for miscoverage rate $\tilde{\alpha} = (\alpha - \delta)/(1 - \delta)$. Then, for any new query $\boldsymbol{x}_{new} \in \mathcal{X}$,*

$$\mathbb{P}\big(f(\boldsymbol{x}_{new}, \boldsymbol{\lambda}^{\boldsymbol{x}_{new}}) < q + \epsilon\big) \geq 1 - \alpha\,.$$

The proof is deferred to Appendix B. As a consequence of this result, a conformal predictor learned on the noisy data with modified miscoverage rate $\tilde{\alpha}$ can be turned into a valid predictor (with miscoverage rate $\alpha$) for the ground-truth data by increasing the learned rejection threshold by $\epsilon$, provided the bounded noise property (11) can be ascertained. Thus, if we denote the corresponding credal set predictor by $h_\epsilon$, we can guarantee that $\mathbb{P}\big(\boldsymbol{\lambda}^{\boldsymbol{x}_{new}} \in h_\epsilon(\boldsymbol{x}_{new})\big) \geq 1 - \alpha$. For a comprehensive study of handling noisy data in conformal prediction, we refer to [19].

## 4 Experiments

For the sake of comparison, we examine different nonconformity functions in our experiments. When utilizing a first-order predictor, besides total variation (**TV**) and the First Wasserstein (**WS**)

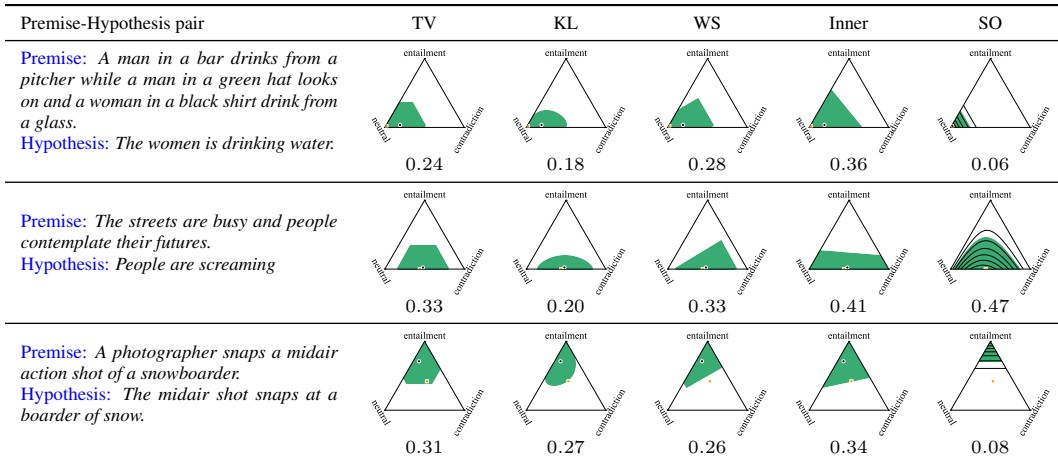

| Premise-Hypothesis pair | TV | KL | WS | Inner | SO |
|---|---|---|---|---|---|
| Premise: *A man in a bar drinks from a pitcher while a man in a green hat looks on and a woman in a black shirt drink from a glass.* Hypothesis: *The women is drinking water.* | 0.24 | 0.18 | 0.28 | 0.36 | 0.06 |
| Premise: *The streets are busy and people contemplate their futures.* Hypothesis: *People are screaming* | 0.33 | 0.20 | 0.33 | 0.41 | 0.47 |
| Premise: *A photographer snaps a midair action shot of a snowboarder.* Hypothesis: *The midair shot snaps at a boarder of snow.* | 0.31 | 0.27 | 0.26 | 0.34 | 0.08 |

Figure 2: Various credal sets obtained for three instances from ChaosNLI dataset. The ground truth distributions are denoted by orange squares. Black circles indicate model predictions in cases employing a first-order learner (first four columns). For the last column, utilizing a second-order learner, the predicted second-order distributions are represented through contour plots. The miscoverage rate is $\alpha = 0.2$, and the efficiency of each credal set is depicted below it.

distance, we also investigate the Kullback–Leibler (**KL**) divergence and 1 minus the inner product (**Inner**) as nonconformity functions. For the second-order predictor, we consider 1 minus the relative likelihood (**SO**) as defined in (8). Furthermore, we demonstrate how our credal sets allow uncertainty quantification and the disentanglement of total uncertainty into epistemic and aleatoric components. We focus on a measure proposed by Abellán et al. [2], which regards the upper Shannon entropy $H$[1] of a given credal set $Q$ as total uncertainty, the lower Shannon entropy as aleatoric, and the difference between upper and lower entropy as epistemic:

$$\underbrace{H^*(Q)}_{\text{total}} = \underbrace{H_*(Q)}_{\text{aleatoric}} + \underbrace{(H^*(Q) - H_*(Q))}_{\text{epistemic}} \quad \text{with } H^*(Q) := \max_{\boldsymbol{\lambda} \in Q} H(\boldsymbol{\lambda}), \ H_*(Q) := \min_{\boldsymbol{\lambda} \in Q} H(\boldsymbol{\lambda}). \quad (12)$$

We refer to [23] for other measures, along with a discussion of their corresponding strengths and weaknesses. Interestingly enough, the interval $[H_*(Q), H^*(Q)] \subseteq [0, 1]$ can be seen as an alternative characterization of the credal set $Q$, which helps address the challenge of visualization for $K > 3$.

In this section, we focus on experiments on two real-world datasets. Further information on these datasets and details about the learning models can be found in Appendix C. Additional experiments on synthetic data, including an illustrative example showing how the resulting credal sets change as epistemic uncertainty decreases, and experiments on the impact of imprecise first-order data, are provided in Appendix E. All implementations and experiments can be found on our GitHub repository.[2]

**ChaosNLI Dataset.**   We start our experiments with a highly ambiguous dataset, ChaosNLI [31], where the task is to classify the textual entailment of a premise-hypothesis pair into three classes: entailment, contradiction, and neutral. We train both first-order and second-order predictors using this data and construct credal sets with all five nonconformity functions to facilitate a comprehensive comparison of these methods. In Figure 2, we compare the resulting credal sets of different nonconformity functions for three specific instances. As expected, even though CP should work regardless of the choice of nonconformity score function, this choice affects the size and geometry of the prediction set. To compare the prediction set size, aka *efficiency*, across different nonconformity functions, we discretize the simplex using a fine grid. The efficiency is gauged by considering the fraction of all distributions that lie within the predicted credal sets. We perform training, calibration, and testing using 10 different random seeds and depict the average coverage and efficiency results of each credal set predictor under different miscoverage rates ($\alpha$) in Figure 3. Notably, the mean of the

---

[1]$H(\boldsymbol{\lambda}) := -\sum_{k=1}^{K} \lambda_k \log_K(\lambda_k)$ with $0 \log 0 = 0$ by definition.
[2]The link to the code: `https://github.com/alireza-javanmardi/conformal-credal-sets`

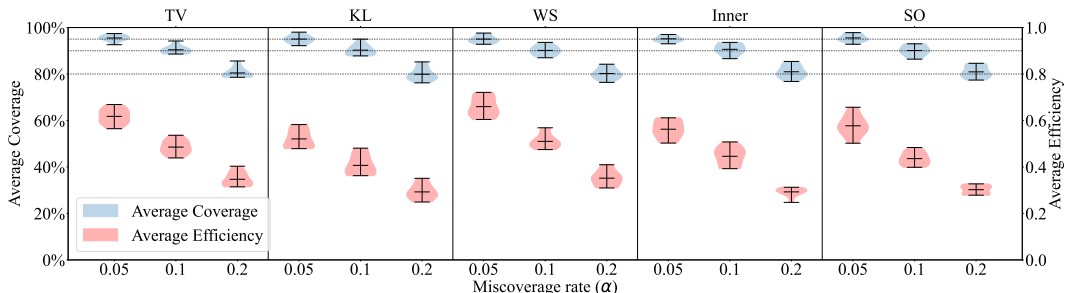

Figure 3: Coverage and efficiency results of different nonconformity functions applied on the ChaosNLI dataset. The horizontal dashed lines indicate the nominal coverage levels.

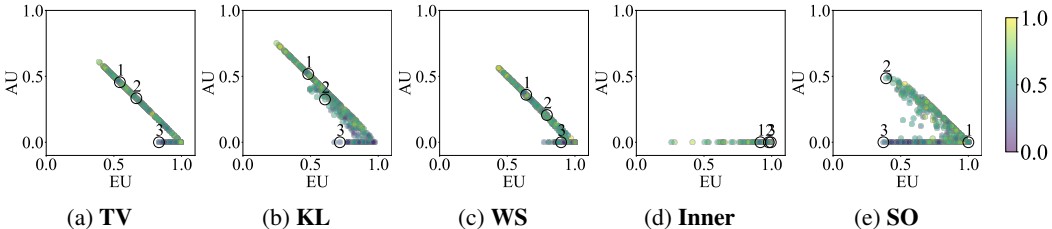

| (a) **TV** | (b) **KL** | (c) **WS** | (d) **Inner** | (e) **SO** |

Figure 4: Scatter plots of aleatoric versus epistemic uncertainty using various credal set predictors for the ChaosNLI dataset with $\alpha = 0.2$. The colors indicate the entropy of the ground truth distribution. The three cases numbered 1 to 3, correspond to the first to third rows of instances shown in Figure 5.

average coverage over the test data across various random seeds aligns with or exceeds the nominal value, consistent with the conformal prediction guarantee.

We calculate the uncertainty intervals as in (12) for all (test) instances. Figure 4 provides scatter plots of quantified AU vs. EU for different credal set predictors, with colors indicating the entropy of the ground-truth distribution. These plots serve as an evaluation of uncertainty quantification performance. Generally, given access to the first-order distribution, we expect the following: if the EU is low, the AU should align closely with the entropy of the ground-truth first-order distribution. However, when the EU is high, the quantified AU may vary, appearing either close to or far from this entropy. In contrast, with only standard zero-order data available, such direct evaluation isn't feasible, which is why indirect evaluation methods like OOD detection or accuracy-rejection curves are preferred.

| | Premise | Hypothesis | Q | $[H_*(Q), H^*(Q)]$ |
|---|---|---|---|---|
| **High EU** | *The purpose of the Diwan-i-Khas is hotly disputed; it is not necessarily the hall of private audience that its name implies.* | *The hall is not know many people.* | | |
| **Low EU, High AU** | *For example, Bruce Barton's The Man Nobody Knows , a best seller in 1925-26, portrays Jesus as the ultimate business-man.* | *Bruce Barton's, "The Man Nobody Knows", a best seller in 1925-26, is known as the best exam-ple of Jesus as the ulti-mate businessman.* | | |
| **Low EU, Low AU** | *A woman in a long-sleeved shirt checks her phone as a man in a leather jacket passes be-hind her.* | *The man is passing by on his way to the store.* | | |

Figure 5: Different uncertainty situations given the predicted credal sets generated by **SO** method for the ChaosNLI dataset with $\alpha = 0.2$.

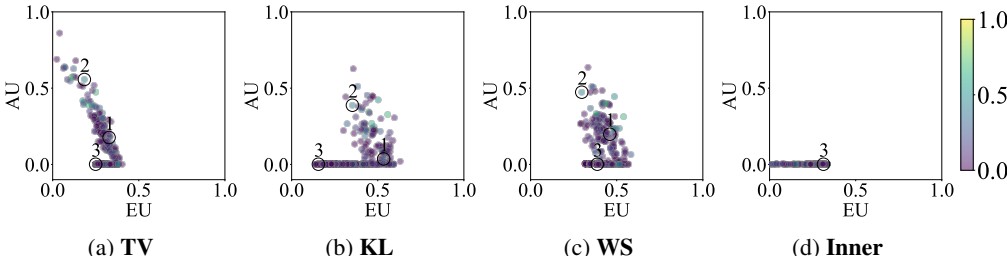

|     (a) **TV**     |     (b) **KL**     |     (c) **WS**     |     (d) **Inner**     |

Figure 6: Scatter plots of aleatoric versus epistemic uncertainty using various credal set predictors for the CIFAR10-H dataset with $\alpha = 0.2$. The colors indicate the entropy of the ground truth distribution. The three cases numbered 1 to 3, correspond to the first to third rows of instances shown in Figure 7.

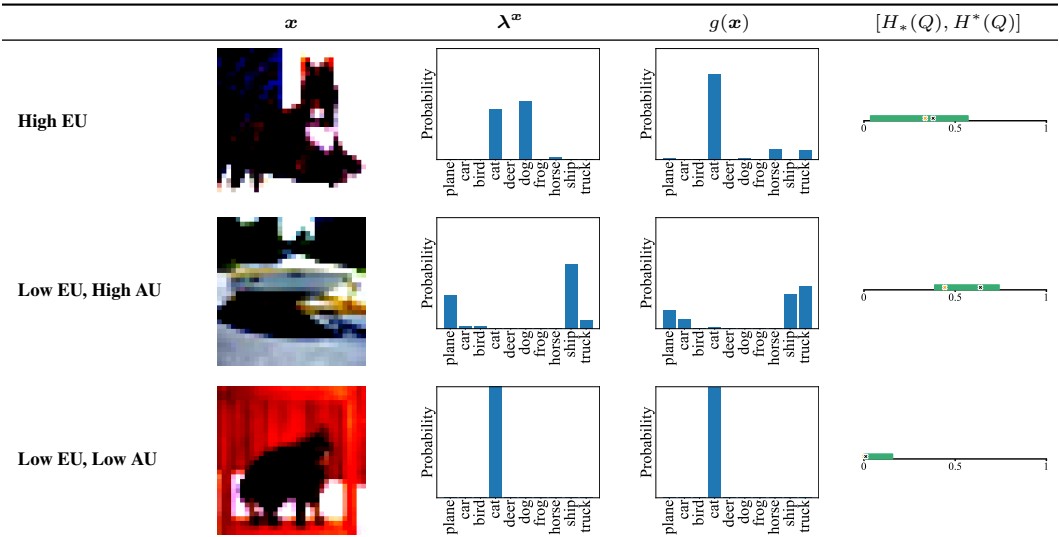

Figure 7: Different uncertainty situations given the predicted credal sets generated by **KL** method for the CIFAR10-H dataset with $\alpha = 0.2$.

For the **SO** method, three uncertainty cases are chosen from the results: high EU, low EU with high AU, and low EU with low AU as shown in Figure 5. When visualizing the uncertainty intervals, we display the entropy of the ground truth distribution in orange and the predicted entropy for the first-order predictor or the mean of the Dirichlet distribution for the second-order predictor in black. However, note that covering the entropy of a distribution with the uncertainty interval does not equate to covering the distribution itself with the credal set.

**CIFAR-10H Dataset.** We also apply our methods to the CIFAR-10H dataset [35], which contains the distributions on the classes for the CIFAR-10 test set derived from human annotations. This time, we simply use a first-order predictor pre-trained on the CIFAR-10 training set (i.e., standard zero-order training data) and only use CIFAR-10H for calibration and testing. Since the visualization of credal sets is no longer possible for this dataset, we limit our attention to uncertainty quantification analysis. Figure 6 illustrates scatter plots of quantified AU versus EU for different credal set predictors. Three uncertainty cases are chosen from the results and shown in Figure 7. For instance, in the first row, the model's predicted distribution ($g(\boldsymbol{x})$) heavily favors "cat" over "dog", while human-derived distributions assign nearly equal probabilities to both classes. This relatively high epistemic uncertainty is effectively captured by the uncertainty interval.

**Further Discussions.** Given a model and a data point, different methods may exhibit different behaviors in terms of uncertainty representation, stemming from their distinct approaches to generating the sets. For instance, it has been observed that **KL** tends to limit the set's expansion towards high entropy regions compared to other methods. The **Inner** method, on the other hand, tends to construct

credal sets by incorporating corners (degenerate distributions), thereby setting the lower entropy to zero. Consequently, it may fail to effectively represent different uncertainty components in a reliable manner. Moreover, this method has the potential to yield empty credal sets, meaning that even the prediction of the first-order predictor may be excluded from the credal set.

From an uncertainty quantification perspective and considering the measure in (12), it becomes apparent that set size may not always accurately reflect epistemic uncertainty. In fact, a credal set of a certain size positioned in the middle of simplex may contain less epistemic uncertainty than the same set located around a corner.

The observed dependency between quantified aleatoric and epistemic uncertainty in Figures 4 and 6 is partly due to the chosen measure in (12): Generally, higher epistemic uncertainty implies a larger uncertainty interval, resulting in a lower value for aleatoric uncertainty. Thus, this dependency is inevitable in the high EU region. Another unintended behavior observed in the **TV**, **KL**, and **WS** approaches is that low epistemic uncertainty coincides *solely* with high aleatoric uncertainty. This issue arises mainly from the lack of adaptivity in the credal sets constructed by first-order predictors. In Appendix D, we propose a method to enhance the adaptivity of these approaches using the concept of normalized nonconformity functions [33].

## 5   Limitations

The methods we propose are promising but still subject to certain limitations. For instance, for our methods to perform effectively, we require first-order data, at least for calibration. While such data is becoming increasingly available in practice, it is not accessible for all datasets and domains. Besides, for our generalization to the case of imprecise first-order data, practical implementation depends on a meaningful choice of the hyperparameters $\epsilon$ and $\delta$ to ensure inference that is both valid and efficient. When labels are based on relative frequencies (as in the case of multiple annotators), classical statistical methods might apply. However, determining an appropriate choice of $\epsilon$ and $\delta$ for broader practical problems remains an open issue. Another challenge lies in representing credal sets as subsets of the probability simplex. Although the credal sets can always be precisely described by equation (10), for the nonconformity functions used in this work, there are no closed-form equations for the resulting credal sets. Instead, the sets are represented implicitly through the nonconformity threshold. Numerical approximation is feasible but generally limited to scenarios with a small number of classes. The representation issue is also connected to computing uncertainty measures for quantifying epistemic and aleatoric uncertainty in a credal set that involves the computation of specific set characteristics [26, 41].

## 6   Conclusion and Future Work

Conformal credal set prediction connects machine learning with imprecise probability theory and offers a novel data-driven approach to constructing predictions that effectively capture both aleatoric and epistemic uncertainty. Thereby, it provides the basis of a new approach to reliable, uncertainty-aware machine learning. Leveraging the inherent validity of the conformal prediction framework, our conformalized credal sets are assured to cover the ground truth distributions with high probability. We have explored different nonconformity functions within this novel setting and evaluated their performance through numerical experiments.

There are several promising directions for future work. One avenue is to extend our method to standard (zero-order) data. While it has been shown that learning a second-order predictor from such data is challenging [8, 9], whether similar problems apply to credal predictors constructed from such data is not yet fully clear. Additionally, exploring other nonconformity functions that lead to closed-form solutions for credal sets or enhance efficiency and reduce uncertainty is worth considering. Finally, constructing a set of labels from our proposed credal sets presents an intriguing opportunity for further research, especially as such sets can provide more information compared to standard conformal prediction sets.

**The broader impact** of our work is the advancement of Machine Learning models towards better uncertainty-aware predictions, and we do not foresee any negative societal impacts.

## Acknowledgment

Alireza Javanmardi was supported by the Deutsche Forschungsgemeinschaft (DFG, German Research Foundation): Project number 451737409.

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

# Appendix A: Extended Related Work

**Credal sets** are widely used as models for representing uncertainty, notably within the domain of imprecise probabilities [58]. As already mentioned, they can represent both types of uncertainty, aleatoric and epistemic. In the context of data analysis and statistical inference, credal sets are often used as robust models of prior information, namely for modeling imprecise information about the prior in Bayesian inference [59].

In machine learning, credal sets have been used for generalizing some of the standard methods, including naive Bayes [14, 62], Bayesian networks [15], and decision trees [1]. Typically, these approaches generalize simple frequentist inference to robust Bayesian inference, making use of an imprecise version of the Dirichlet model (a conjugate prior for the multinomial distribution). Compared to our approach, these methods are learning on standard (zero-order) training data. Moreover, despite representing uncertainty in predictions, they do not provide any formal guarantees.

**Conformal prediction** [57], briefly introduced in Section 2.2, has recently gained attention for various applications in machine learning, especially for classification tasks [4, 20, 38, 40, 50]. These methods mostly focus on split conformal prediction using a held-out calibration set [32], overcoming computational limitations of earlier transductive or bagging approaches [7, 30, 49, 55, 57]. While tackling classification tasks, our method for constructing conformal credal sets has more similarity with conformal regression [37, 45], particularly in multivariate settings [17], as we essentially conformalize the simplex space of categorical distributions. Our nonconformity scores differ, however, in that they are specific for distributions rather than considering general multivariate spaces. This work also relates to work on appropriate measures of inefficiency [56] as measuring the inefficiency of our conformal credal sets is non-trivial. Most closely related to our work is the recent work by Stutz et al. [52], who consider conformal prediction in settings with high aleatoric uncertainty. However, we explicitly target the construction of conformal credal sets, while Stutz et al. [52] mainly focus on constructing confidence sets of classes. The connection between CP and credal sets has also been explored by Cella and Martin [13] and Lienen et al. [29]. However, their emphasis lies on standard (zero-order) data, which fails to represent uncertainty truthfully. Furthermore, the sets generated by their methods are consistently confined to the simplex corners (around degenerate distributions) and display notably conservative behavior.

**First-order data.** In settings with high aleatoric uncertainty, labeling each example with a single, unique class is clearly insufficient. In practice, this is typically captured by high disagreement among annotators – a problem particularly common in natural language tasks [3, 5, 6, 18, 34, 36, 39, 42] or even in computer vision [10, 31, 35, 43, 63]. Handling this disagreement has received considerable attention lately [54] as it offers to go beyond this zero-order information. For example, recent work on evaluation with disagreeing annotators [51] argues the use of these annotations to get approximate first-order information for evaluation. This approach is becoming more and more viable with crowdsourcing tools [27, 47, 48] being an integral component of the benchmark, making multiple annotations per data instance more accessible. We follow a similar approach in our construction of conformal credal sets.

# Appendix B: Proof of Theorem 3.2

*Proof.* Let $A$ denote the event $f(\boldsymbol{x}_{\text{new}}, \boldsymbol{\lambda}^{\boldsymbol{x}_{\text{new}}}) < q + \epsilon$ and $\tilde{A}$ the event $f(\boldsymbol{x}_{\text{new}}, \tilde{\boldsymbol{\lambda}}^{\boldsymbol{x}_{\text{new}}}) < q$. We have

$$P(A) \geq P(A \wedge \tilde{A})$$
$$= P(\tilde{A}) \cdot P(A \mid \tilde{A})$$
$$= P(\tilde{A}) \cdot (1 - P(\neg A \mid \tilde{A}))$$

Since $\neg A$ means that $f(\boldsymbol{x}_{\text{new}}, \boldsymbol{\lambda}^{\boldsymbol{x}_{\text{new}}}) \geq q + \epsilon$, the conditional event $\neg A \mid B$ implies a violation of the closeness condition in (11), wherefore the probability $P(\neg A \mid B)$ is upper-bounded by $\delta$ according to (11). Therefore, noting that $P(\tilde{A}) \geq 1 - \tilde{\alpha}$ is the standard guarantee by CP,

$$P(A) \geq P(\tilde{A}) \cdot (1 - P(\neg A \mid \tilde{A}))$$
$$\geq (1 - \tilde{\alpha}) \cdot (1 - \delta)$$
$$= 1 - \alpha.$$

$\square$

## Appendix C:   Real-World Datasets and Models Overview

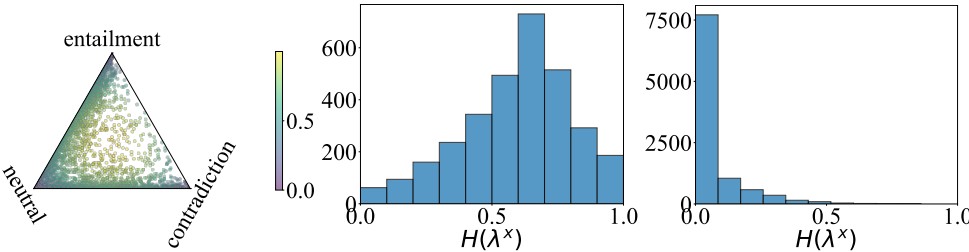

(a) Scatter plot of the ChaosNLI distributions.

(b) Histogram of the entropies of ChaosNLI distributions.

(c) Histogram of the entropies of CIFAR10-H distributions.

Figure 8: General overview of real-world datasets.

**ChaosNLI** [31] (License: CC BY-NC 4.0 DEED) is an English Natural Language Inference (NLI) dataset that captures the inherent variability in human judgments of textual entailment. Here, the classes are *entailment*, *neutral*, and *contradiction* for each premise-hypothesis pair. Instances in this dataset are selected from the development sets of SNLI [12], MNLI [60], and AbductiveNLI [11], for which the majority vote was less than three among the five human annotators. These instances were then given to $100$ independent humans for annotation, given strict annotation guidelines. We combine the chaos-SNLI and chaos-MNLI subsets, resulting in a dataset of $3113$ datapoints. For model training, we leverage a language model from the Hugging Face `transformers` library [61], initially trained on SNLI and MultiNLI datasets for classification tasks[3]. We utilize the last hidden layer output of this model to embed the premise-hypothesis pairs from all $3113$ instances into $768$-dimensional vectors, serving as inputs for the learning model. To split the data, we randomly select $500$ instances for calibration, $500$ for testing, and the remaining for training.

As for the learner, we employ a deep neural network consisting of three hidden layers with $256$, $64$, and $16$ units, utilizing ReLU as the activation function. Prior to the output layer, a dropout layer with a rate of $0.3$ is incorporated. The same model architecture serves both first- and second-order predictors, differing only in the activation functions of the output layers. For the first-order predictor, softmax is used, while for the second-order predictor, ReLU is employed. Learning is facilitated using the Adam optimizer with a learning rate of $10^{-4}$, utilizing cross-entropy as the loss function for the first-order predictor and negative log-likelihood for the second-order predictor.

**CIFAR10-H** [35] (License: CC BY-NC-SA 4.0 DEED) is a dataset of soft labels that capture human perceptual uncertainty for the $10000$ images of CIFAR-10 test set [28]. A total of $511,400$ human classifications were gathered via Amazon Mechanical Turk, excluding participants who performed poorly on obvious images. On average, each image received $51$ judgments, with the number of judgments per image ranging from $47$ to $63$. As shown in Figure 8, the histogram of the entropies of CIFAR-10H distributions shows that this dataset is less ambiguous compared to ChaosNLI. For this dataset, we don't perform any model training and instead use a model pre-trained on the CIFAR-10 training images[4]. We utilize the model's predicted distributions along with the CIFAR-10H distributions to construct credal sets, focusing solely on first-order approaches. We randomly select $1000$ instances for testing and the rest for calibration.

## Appendix D:   Enhancing Adaptivity in First-Order Predictor Credal Sets

As mentioned earlier, with methods based on the first-order predictor, once the quantile is determined during calibration, all distributions within a certain distance are included in the set for any given point at prediction time. This indicates that these methods lack adaptivity and do not account for the local heterogeneity of the data. Given the uncertainty measure in (12), for a given quantile or

---

[3]The model can be found at https://huggingface.co/cross-encoder/nli-deberta-base with Apache License 2.0

[4]The pre-trained model can be found at https://github.com/huyvnphan/PyTorch_CIFAR10 with MIT license.

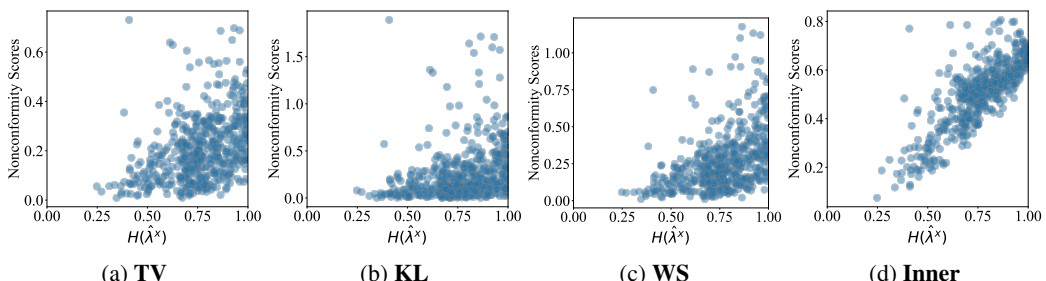

|             |             |             |              |
|:-----------:|:-----------:|:-----------:|:------------:|
| (a) **TV**  | (b) **KL**  | (c) **WS**  | (d) **Inner**|

Figure 9: Scatter plots of calibration scores versus the entropy of the first-order predictor's output for different nonconformity scores for the ChaosNLI dataset.

set size, when the model's predicted distribution is near the simplex corner, the difference between the highest and lowest entropies is much greater than when the predicted distribution is located in the middle. That is why, in Figure 4, small epistemic uncertainty is observed only for high aleatoric cases for methods based on the first-order predictor. To handle this issue, one approach involves utilizing the normalized nonconformity score by dividing the score of each point by (a measure of) its dispersion at that point [33]. As depicted in Figure 9, we've noticed a direct correlation between the score's dispersion and the entropy of the predicted distributions for the ChaosNLI dataset; specifically, higher entropy corresponds to greater variability in the scores. Therefore, we propose the following normalized score:

$$\tilde{f}_1(\boldsymbol{x}, \boldsymbol{\lambda}^{\boldsymbol{x}}) := \frac{f_1(\boldsymbol{x}, \boldsymbol{\lambda}^{\boldsymbol{x}})}{H(\hat{\boldsymbol{\lambda}}^{\boldsymbol{x}}) + \tau} \,, \tag{13}$$

where $H(\hat{\boldsymbol{\lambda}}^{\boldsymbol{x}})$ denotes the entropy of the predicted distribution at point $\boldsymbol{x}$ and $\tau$ is a hyperparameter that prevents division by zero and controls the influence of $H(\hat{\boldsymbol{\lambda}}^{\boldsymbol{x}})$ on the scores (the higher the $\tau$, the lower the influence). The credal set for $\boldsymbol{x}_{\text{new}}$ will be constructed as

$$\left\{ \boldsymbol{\lambda} \in \Delta^K \mid f_1(\boldsymbol{x}_{\text{new}}, \boldsymbol{\lambda}) \leq q(\tilde{\mathcal{E}}_1, \alpha')(H(\hat{\boldsymbol{\lambda}}^{\boldsymbol{x}_{\text{new}}}) + \tau) \right\},$$

where $\tilde{\mathcal{E}}_1$ is the set of normalized nonconformity scores. Figure 10 compares the scatter plots of AU versus EU for all first-predictor-based models with their adaptive counterparts with $\tau = 0.1$. The adaptive approach decreases the average epistemic uncertainty. For example, the positions of the three instances of Figure 5 moved closer to their corresponding values of the **SO** method.

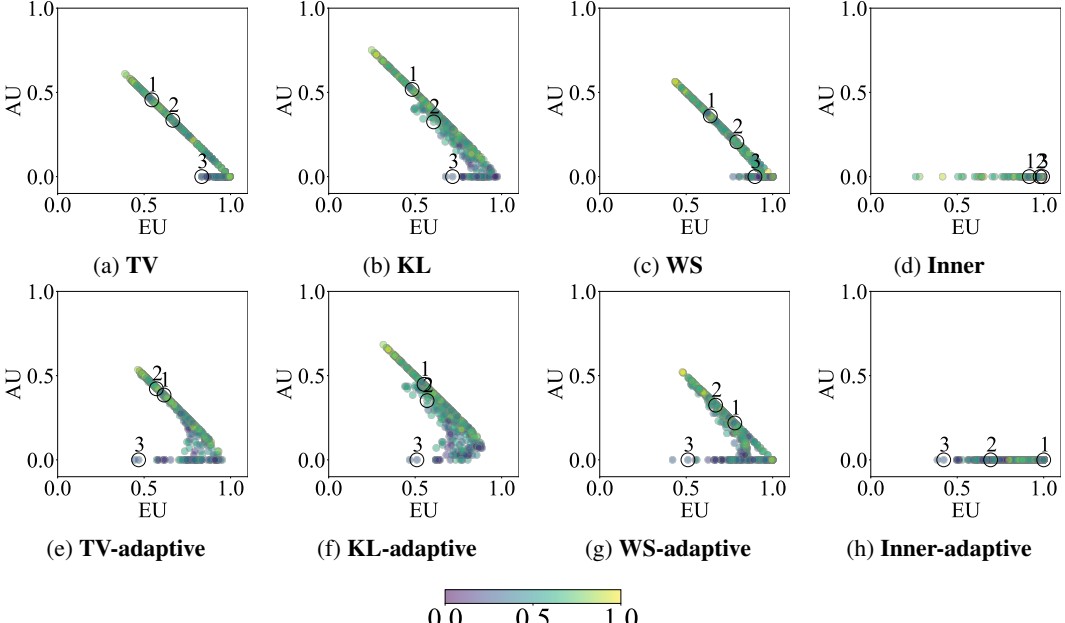

Figure 10: Scatter plots of aleatoric versus epistemic uncertainty using various credal set predictors for the ChaosNLI dataset with $\alpha = 0.2$. The colors indicate the entropy of the ground truth distribution. The three cases numbered 1 to 3, correspond to the first to third rows of instances shown in Figure 5.

## Appendix E:    Synthetic Data

Throughout this paper, we generate synthetic data for a $K$-class classification task as follows: we consider $d$-dimensional features $X \in \mathbb{R}^d$, where $X_1, \ldots, X_d$ are independent standard normal random variables. Subsequently, we generate a random matrix $\beta \in \mathbb{R}^{d \times K}$, with its elements drawn independently from the standard normal distribution. To define the ground truth probability over the classes for object $X$, we use the following formulation:

$$\lambda_k^{\boldsymbol{x}} := \mathbb{P}(Y = k | \boldsymbol{x}) = \frac{Z_k(\boldsymbol{x})}{\sum_j Z_j(\boldsymbol{x})}, \tag{14}$$

where $Z(\boldsymbol{x}) := \exp(\boldsymbol{x}^\top \beta)$. As for the learner (both first-order and second-order predictors), in the subsequent experiments, we utilize the same deep neural network described in Appendix C for the ChaosNLI dataset, with the input layer adopting the d-dimensional feature vector.

### E.1    Illustrative Example

To illustrate the primary concept of credal sets and uncertainty decomposition, we present a straight-forward yet intuitive example. Given the data-generating process in (14), we generate 3000 samples with $d = 10$ and $K = 3$ solely to facilitate the visualization of the credal sets, splitting them equally between training, calibration, and testing datasets. Initially, we train a model with only 10 points, gradually increasing to 50, 100, 500 and 1000. At each stage, we calibrate the model using all the calibration data points and evaluate its performance on the test points. From the test data, we select three examples characterized by high entropy (almost uniform distribution), relatively high entropy (uniform across two classes), and low entropy (almost a Dirac distribution). We then plot their credal sets alongside lower and upper entropies for varying numbers of training data points.

Figure 11 provides intriguing insights! As evident in the first column, where the model was trained with only 10 data points, there exists a significant epistemic uncertainty across all three cases. This indicates that none of the predictions are reliable, and it's challenging to predict a single label for any of the given examples. Moving to the third column, we observe a significant reduction in epistemic uncertainties. In the second example (second row), it indicates that the total uncertainty is primarily due to epistemic uncertainty, whereas for the other two cases, it's a result of high aleatoric uncertainty with some epistemic uncertainty. This suggests that gathering more data (and reducing the total uncertainty) may not significantly enhance the capability to predict a single label for the latter cases. Figure 12 illustrates the evolution of scatter plots of AU versus EU for all first-predictor methods as the number of training data increases.

### E.2    Experiments with Imprecise First-Order Data

We considered another set of experiments with synthetic data to illustrate the impact of imprecise first-order data, particularly to showcase the behavior of the proposed credal sets when we only have access to an approximation of the ground truth distributions. Our experiment revolves around a $K$-class classification task with $K \in \{3, 4, 6, 8, 10\}$. For each $K$, we generate $N = 1500$ samples with $d = 10$ using (14) to construct the datasets $\mathcal{D}^K = \{(\boldsymbol{x}_i, \boldsymbol{\lambda}^{\boldsymbol{x}_i})\}_{i=1}^N$. To obtain imprecise versions of $\mathcal{D}^K$, we employ a sampling approach. Specifically, we independently sample each distribution $\boldsymbol{\lambda}^{\boldsymbol{x}_i}$ $m$ times and utilize relative frequencies to create its noisy counterpart $\tilde{\boldsymbol{\lambda}}_m^{\boldsymbol{x}_i}$. We represent the resulting dataset as $\mathcal{D}_m^K = \{(\boldsymbol{x}_i, \tilde{\boldsymbol{\lambda}}_m^{\boldsymbol{x}_i})\}_{i=1}^N$. We repeat this process four times with $m \in \{1, 5, 10, 100\}$.

Given each dataset $\mathcal{D}_m^K$, we randomly partition data points into 1300 training, 100 calibration, and 100 test instances and perform the proposed methodologies accordingly. Again, we repeat this process ten times with different random seeds for each dataset $\mathcal{D}_m^K$. Due to the computational complexity in calculating efficiency for cases with $K > 3$, we utilize the quantile of the calibration nonconformity scores as an efficiency metric. In Figure 13, we represent the overall result under different $K$ and $m$ values. It can be observed that the coverage is fulfilled across almost all scenarios, including $m = 1$ with degenerate distributions. This observed behavior is somewhat intuitive. The model endeavors to learn the underlying probabilistic relationship between $X$ and $Y$, even given the noisy data [5]. Consequently, during calibration with noisy instances, the nonconformity scores of noise-free

---

[5] Of course, this holds under some reasonable assumptions on noise.

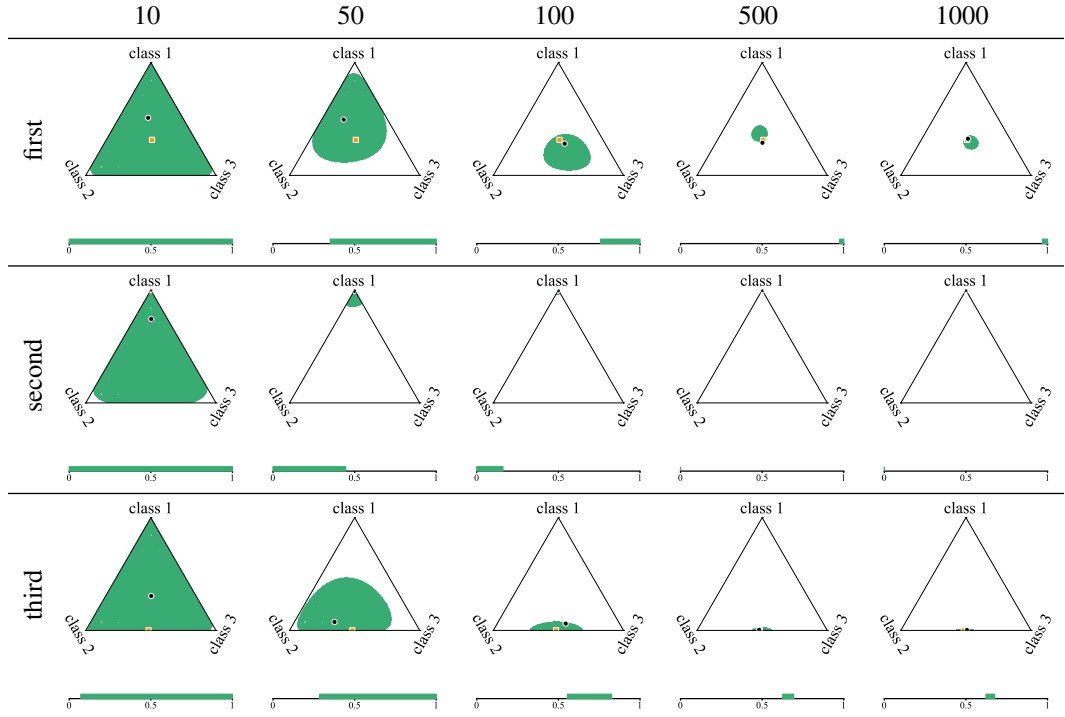

Figure 11: Credal sets generated by the **KL** method for three examples of illustrative synthetic data with $\alpha = 0.1$. Each row represents one of three examples, with columns illustrating the evolution of the predicted credal set based on varying numbers of training data points. The lower and upper entropy of each credal set is displayed below it.

instances are mostly upper-bounded by the scores of their noisy counterparts, resulting in more conservative sets that effectively cover the ground-truth distributions. It can also be seen that the quantile of the nonconformity scores shrinks as $m$ increases. In Figure 14, we illustrate the evolution of the credal sets as $m$ changes from 1 to 100 for different nonconformity functions when $K = 3$. For this case, the full comparison of efficiency and coverage across various nonconformity functions is provided in Figure 15.

## Appendix F:   Experiments Compute Resources

For all experiments, we used an Intel(R) Core(TM) i7-11800H CPU with 16.0 GB of RAM. Model training and calibration steps are quite fast and take only a few minutes to run. In constructing the credal sets, we use high-resolution simplex discretization and determine whether each distribution from the discretized simplex belongs to the credal set via exhaustive search. For the **WS** method, this process can take up to a few seconds per data instance, while for all other methods, it takes less than a second. The uncertainty quantification was also completed in less than a second.

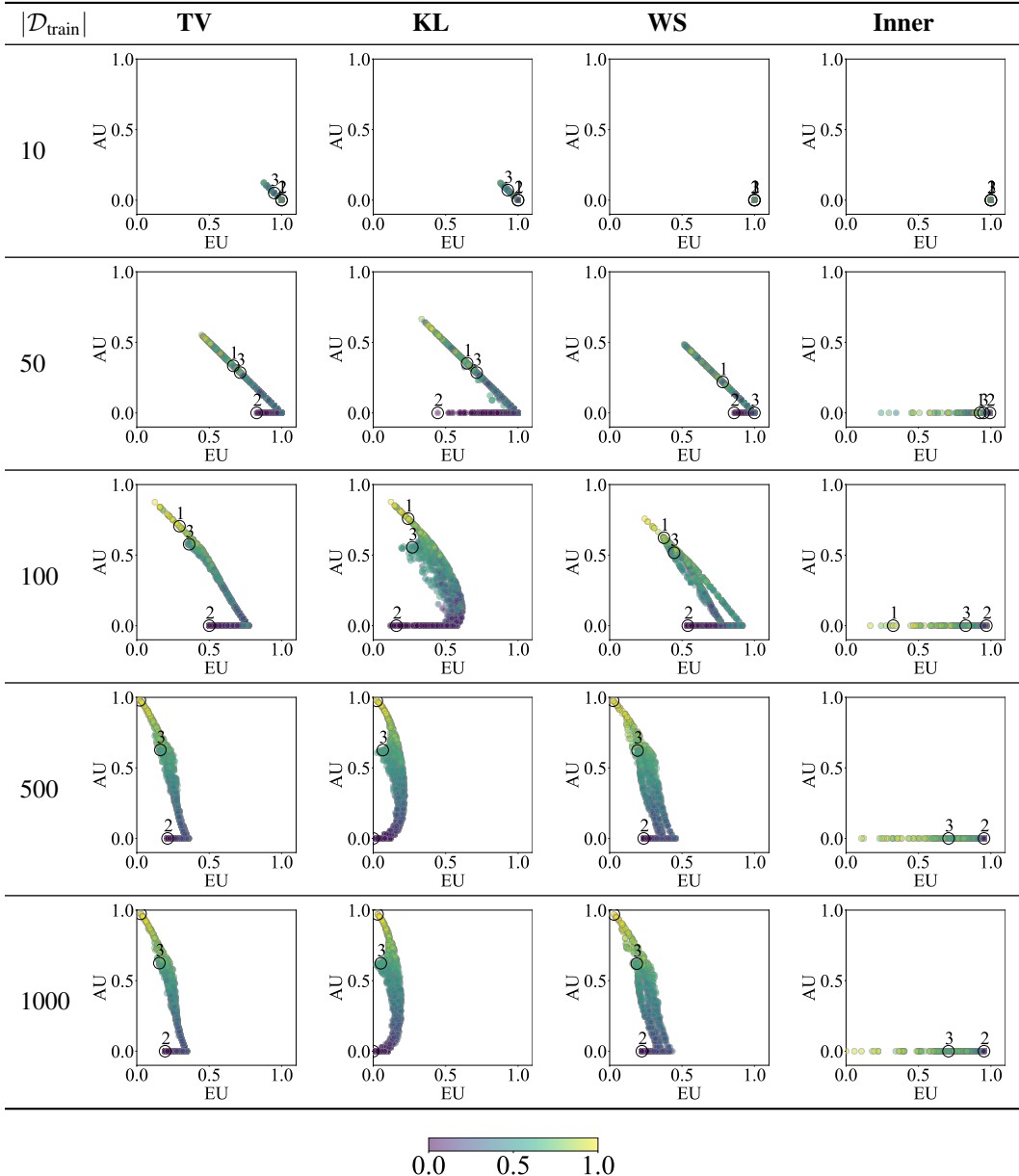

Figure 12: Scatter plots of aleatoric versus epistemic uncertainty using various credal set predictors given different numbers of training data for illustrative synthetic data with $\alpha = 0.1$. The colors indicate the entropy of the ground truth distribution. The three cases, numbered 1 to 3, correspond to the first to third rows of instances shown in Figure 11.

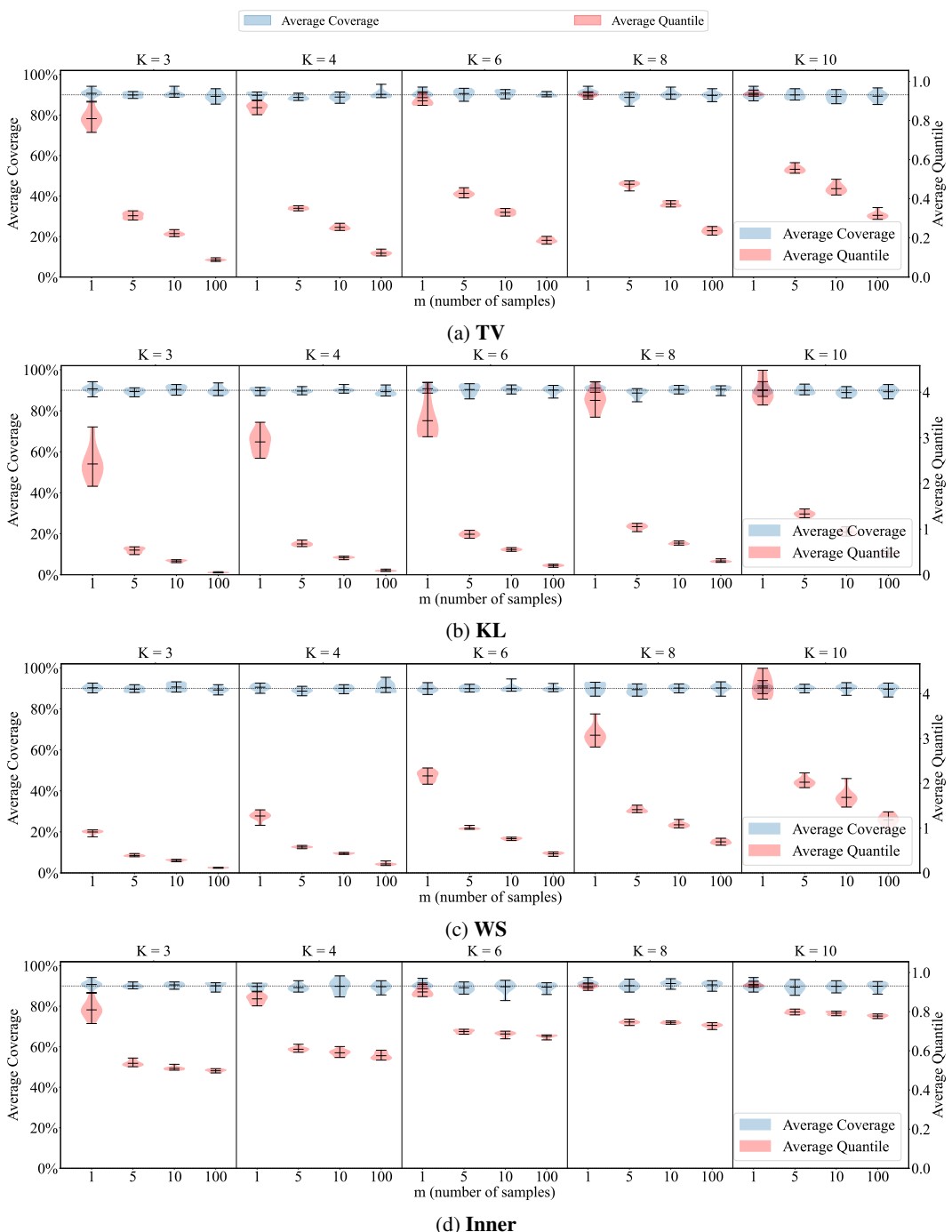

Figure 13: Coverage and quantile results for synthetic data with imprecise first-order distributions, where the ground truth distributions are approximated by observing $m$ samples from them. The horizontal dashed lines indicate the nominal coverage level $1 - \alpha = 0.9$.

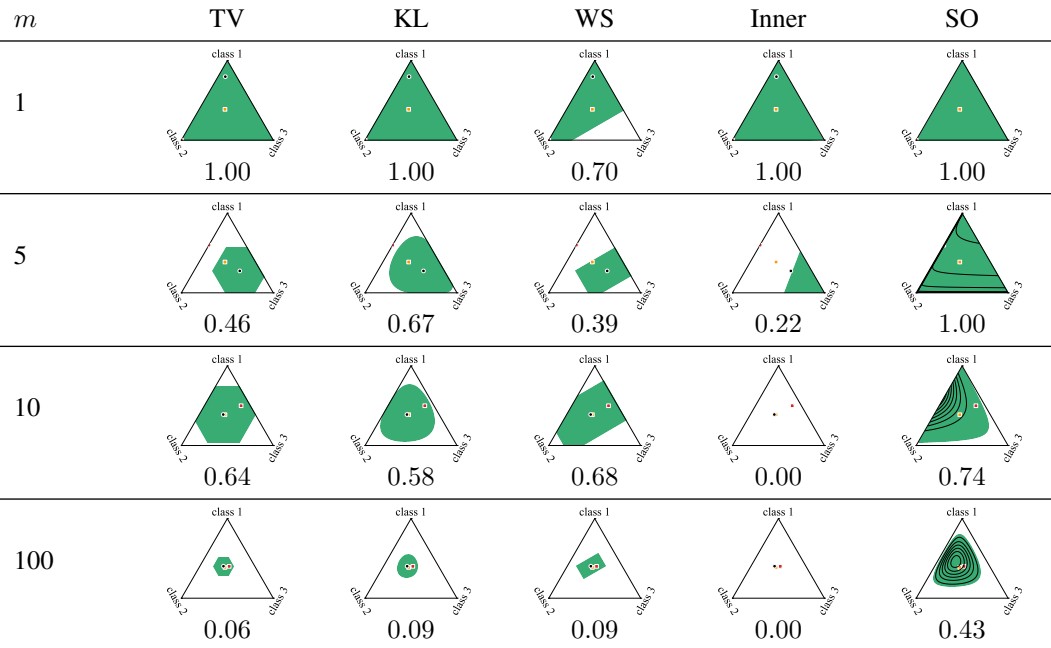

Figure 14: Credal sets derived using various credal set predictors for an example from synthetic data with imprecise first-order distributions. Rows correspond to the number of samples utilized for distribution estimation. The ground truth distribution is marked by an orange square, and its noisy versions are denoted by red squares. In cases employing a first-order learner (first four columns), model predictions are denoted by black circles. The predicted second-order distributions are illustrated via contour plots in the last column, where a second-order learner is employed. The miscoverage rate is $\alpha = 0.05$, and the efficiency of each credal set is indicated below it.

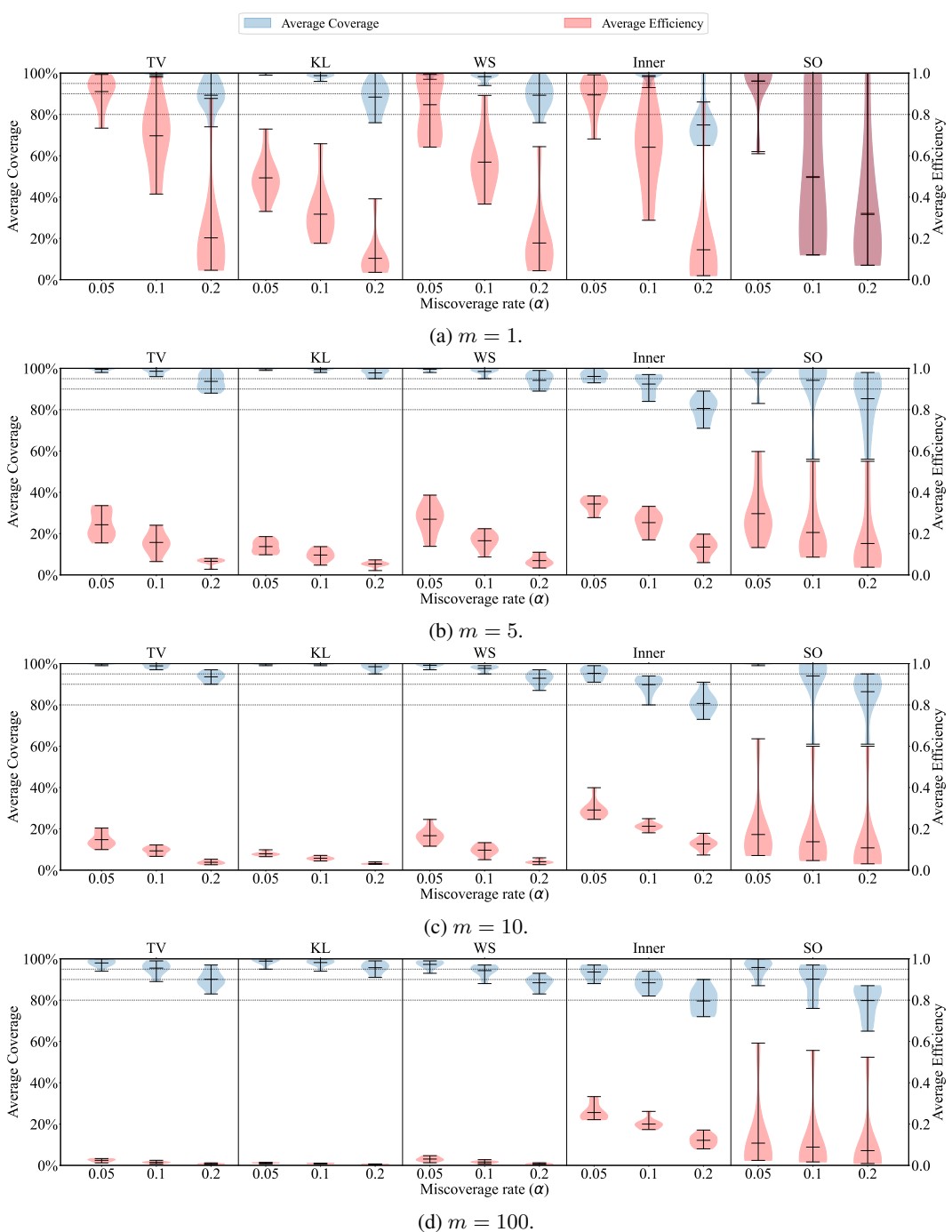

(a) $m = 1$.

(b) $m = 5$.

(c) $m = 10$.

(d) $m = 100$.

Figure 15: Coverage and efficiency results of different nonconformity functions applied on the synthetic data with imprecise first-order distributions ($K = 3$). The horizontal dashed lines indicate the nominal coverage levels.

