# OpenReview forum: "Conformalized Credal Set Predictors"
_NeurIPS.cc/2024/Conference — NeurIPS 2024 poster_

### Official Review · Reviewer_ieV3 · 2024-07-04

**Soundness:** 3
**Presentation:** 4
**Contribution:** 2
**Rating:** 5
**Confidence:** 4

**Summary:**

This paper proposes a method for predicting credal sets in classification tasks, generating conformal credal sets that are guaranteed to be valid with high probability without any assumptions on the model or distribution. The method is applicable to both first- and second-order predictors. Experiments on ChaosNLI and CIFAR10-H demonstrate its effectiveness and how these credal sets allow for practical uncertainty quantification in AU and EU.

**Strengths:**

1. The paper is well-written, with smooth and clear demonstrations. Despite containing many formulas, it is easy to understand.
2. The proposed method combines imprecise probability and conformal prediction to leverage the advantages of both, which is interesting.
3. The experimental design on small label size datasets is comprehensive.
4. The limitations are discussed and soundly pointed out.

**Weaknesses:**

1. The technical novelty is moderate: the proposed method is very similar to a previous conformal prediction method on ambiguous labels [1], as both assume reasonable probability labeling and train a standard (first-order) probability predictor (i.e., the score function). Besides, obtaining a probability interval is technically not very novel.

2. The computational complexity limits the general application of the method to large-scale datasets.

3. Access to ground-truth probability labeling seems unreasonable, especially when model calibration itself is already a significant challenge.

4. The key bounded noise assumption for the labeling process may not be very practical, as mentioned in the paper.


[1] Stutz, David, et al. "Conformal prediction under ambiguous ground truth." arXiv preprint arXiv:2307.09302 (2023).

**Questions:**

1. Would any model calibration methods provide probabilities that satisfy the bounded noise assumption?

2. While some frequentist methods also built on the first-order probability labeling, they realize it as a deterministic action at every sampling step. Would it be possible to adopt this strategy to avoid the assumption? Would the complexity be acceptable?

**Limitations:**

The limitations include the implicit representation of the credal set, the intractability on large label sizes, and the assumption of access to reasonable probability labelings. The authors have adequately discussed these limitations.

---

> ### Author Rebuttal · Authors · 2024-08-06
>
> Thank you for your kind words about the paper and for your valuable feedback. We hope our response addresses your comments and questions.
>
> - **W1.** The main similarity between our work and [1] lies in the data assumption we make. However, while the authors in [1] focus on obtaining prediction sets (i.e., sets of labels), our work is centered on the development of credal sets (i.e., sets of probability distributions). The distinction is important because credal sets allow for the disentangling of total uncertainty into aleatoric and epistemic components, something that prediction sets in [1] do not achieve.
>
>     In the context of credal sets, our proposed conformalized credal sets are designed to be valid in the sense that they cover the ground-truth first-order distribution with high probability—a property that, to the best of our knowledge, is unique to our approach.
>
>     Moreover, for model training, we do not limit ourselves to first-order probability predictors. We thoroughly discuss both first-order and second-order probability predictors, providing a family of nonconformity scores tailored to each type of predictor to enhance the flexibility of our proposed methods. We also address the challenge of obtaining a valid credal set when only imprecise first-order data is available. Additionally, we empirically demonstrate the effectiveness of our proposed methods for uncertainty quantification with such datasets.
> - **W2.** It is worth noting that our approach to model training and calibration does not introduce any additional complexity compared to existing models. Our credal sets can always be precisely described by equation (10) in the paper for any arbitrary K. This equation also enables efficient membership checking, allowing us to determine whether a distribution belongs to a set. The only limitation arises when seeking an exact closed-form representation of the credal set or a compact form, such as in the 1-dimensional regression problem where we represent the set with an interval. However, this challenge is essentially unavoidable for Conformal Prediction in complex (infinite) output spaces, such as multi-output regression or structured output prediction. We have also clarified the details of our experiment in Appendix F, as mentioned in our response to W2 from reviewer gqZt.
> - **W3.** We would like to highlight that datasets where multiple human experts annotate each data instance x are indeed becoming increasingly common in practical tasks, including widely used benchmarks like ImageNet [2] and CIFAR-10 [3], and in some cases, have even been shown to be necessary [4]. This is especially crucial in the context of natural language processing, where a long history of research emphasizes its importance [5,6] (Please refer to our extended related work in Appendix A for many more references). Aggregating annotator disagreements regarding the label of an instance x into a distribution over labels can be seen as an effective way to represent aleatoric uncertainty. We believe and demonstrate in this paper that this information can also contribute to advancements in uncertainty representation and quantification research.
> - **W4.** Guarantees rely on assumptions, and we believe that our assumption is not stronger than assumptions made for comparable results in the literature. Intuitively, the assumption seems to be not only sufficient but even necessary: If noise is unbounded, the estimations can deviate arbitrarily from the ground-truth distribution, thereby making any guarantee impossible.
> Moreover, it's important to highlight that in real-world applications, quantifying noise, if present, may not always be feasible. Instead, we are often limited in the observations that are available to us. For instance, in our case, we rely on distributions obtained from aggregating human annotations. It's also worth noting that the guarantee of the proposed method remains always consistent with respect to these observations.
> - **Q1.** If the reviewer is referring to model calibration with zero-order data, to the best of our knowledge, none of the existing calibration methods are instance-wise, i.e., they do not aim to calibrate the predicted probability for each specific input x but rather provide calibration in expectation. It is certainly interesting to explore various calibration methods to see if they can achieve this.
> - **Q2.** We are not entirely certain that we fully understand the method the reviewer is referring to. We believe we could provide a more comprehensive answer with additional context. However, as a general remark, if we aim to remove the bounded noise assumption, we would either need to introduce an alternative assumption (which may come naturally with a certain approach) or reconsider our theorem or guarantee entirely.
>
> [1] D. Stutz et al., "Conformal prediction under ambiguous ground truth."
>
> [2] L. Beyer et al., "Are we done with imagenet?"
>
> [3] J. C. Peterson et al., "Human uncertainty makes classification more robust"
>
> [4] L. Schmarje, et al. "Is one annotation enough? a data-centric image classification benchmark for noisy and ambiguous label estimation"
>
> [5] Y. Nie et al., "What can we learn from collective human opinions on natural language inference data?"
>
> [6] D. Stutz et al., "Evaluating AI systems under uncertain ground truth: a case study in dermatology"

---

> > ### Author Response · Authors · 2024-08-12
> >
> > Dear Reviewer ieV3,
> >
> > Regarding W4, we would also like to draw your attention to our response to W2 of Reviewer LLVn, where we explained how the bounded noise assumption translates into the noise arising from estimating the ground truth first-order distributions using only samples from that distribution.
> >
> > Also, given that we have addressed all your questions and concerns, we'd love to know if there are any additional ways to improve our work so that it becomes more convincing and impactful, beyond borderline acceptance, for both you and the broader community in the future.
> >
> > Best regards,
> >
> > The Authors

---

> > > ### Comment · Reviewer_ieV3 · 2024-08-13
> > >
> > > I have carefully read the authors' response and appreciate their efforts in answering my questions. However, I maintain my opinion that the contribution and impact are moderate. Although there are synthetic datasets with probabilistic labeling, I believe the most important application of inherent labeling uncertainty is in the medical domain, which the authors did not address in the experiments. Given that current medical machine learning models often suffer from poor performance, may I ask how this would influence the conformal credal sets?

---

> > > > ### Author Response · Authors · 2024-08-13
> > > >
> > > > Thank you for the feedback.
> > > >
> > > > We would like to highlight that our main experiments and results, presented in the main part of the paper, are conducted on two **real** datasets: **ChaosNLI** and **CIFAR10-H**. We found ChaosNLI particularly intriguing due to the notable disagreement among annotators, indicating high aleatoric uncertainty. Additionally, the feasibility of visualizing the credal sets within a 2-D simplex adds to its appeal. We included CIFAR10-H in our experiments because CIFAR10 is widely used in numerous studies, and we aimed to demonstrate the applicability of our method on such datasets to gain a better understanding of uncertainty. In the medical domain, we are only aware of the Google Dermatology DDX dataset [1], where the task is to classify a skin image into a specific skin disease. However, For this dataset, only the annotations and predictions of a model are available, and unfortunately, the images are not publicly accessible (please see [here](https://github.com/google-deepmind/uncertain_ground_truth)). This limited us to conformalizing that given model, which did not provide sufficient flexibility for our experiments. Nevertheless, we can incorporate this dataset into the camera-ready version of the paper if you find it beneficial.
> > > >
> > > > Regarding synthetic data, we employed it for two specific purposes: first, as an illustrative example to show the behavior of our credal sets constructed with different models as the number of training data increases and epistemic uncertainty decreases to (almost) zero. Second, we used synthetic data to evaluate the performance of our algorithm under the assumption of imprecise first-order data, which is impossible to test with any real dataset. Therefore, we believe that these synthetic data experiments were necessary to further demonstrate the potential of our work.
> > > >
> > > > Regarding the raised question, poorer model performance translates to a greater distance between the true first-order distribution and its prediction for the first-order predictor or to a lower predicted density for the true first-order distribution given the second-order predictor. In either case, this generally results in higher nonconformity scores, which eventually lead to larger credal sets. These larger credal sets represent higher epistemic uncertainty, indicating a greater lack of knowledge.
> > > >
> > > > [1] D. Stutz et al., "Conformal prediction under ambiguous ground truth."

---

### Official Review · Reviewer_LLVn · 2024-07-10

**Soundness:** 2
**Presentation:** 3
**Contribution:** 2
**Rating:** 4
**Confidence:** 4

**Summary:**

Authors propose a new uncertainty estimation method based on credal sets predictors. The new approach is based on the Conformal Prediction framework and shares its finite-sample and distribution-free guarantees. Authors verify the validity of their approach on existing NLP and CV datasets.

**Strengths:**

* theoretical part is based on an established mathematical framework and authors formally prove the validity of their method
* authors provide full (anonymized) source code of their experiments

**Weaknesses:**

* The proposed 2nd order score function seems to be not well- defined. For example, if any of $\theta^x_k$ are small than 1, then maximum in denominator of equation (8) becomes infinite and score function becomes constant. This is undesirable behavior to the best of my understanding.
* While it is straightforward to assess the influence of the noise in terms of perturbations of score function, the characterization of the noise in such terms seems not natural. The most practical noise is the one that is due to the categorical observations of the label distribution. It seems that there is a gap between these two characterizations of noise.
* the ability of the method to clearly separate (“disentangle”?) aleatoric and epistemic uncertainty is not fully understood. Judging by Fig.4,6,12, there is a clear functional dependency and in the main text part 4 we read: “approaches with first-order predictors lack adaptivity” with regard to a similar issue. Perhaps a special synthetic experiment can provide more insight whether it is an issue with the base method or the proposed approach.

**Questions:**

* Can you provide a theoretical result in noisy case in terms of the standard observations, i.e. one class per data point?
* Can you fix an issue with the score function?
* On Figures 4,6,12 the data point collapse to a low-dimensional manifold. It is especially visible on Figure 4, where there is a clear linear dependency between AU and EU at least on a portion of the data. In light of this evidence, how can authors support the claim of part 6: “effectively capture both epistemic and aleatoric uncertainty”?
* Authors introduce “uncertainty interval” as an additional uncertainty measure and report it in their figures. Part 4 also contains the following passage: “covering the entropy of a distribution with the uncertainty interval does not equate to covering the distribution itself with the credal set”. Could you please further explain the meaning and the role of this additional measure and why covering the (predicted) entropy may be desirable?
* Conformal prediction can produce sets of different size depending on the non-conformity measure. Figure 3 reports the average set sizes as efficiency for the selected non-conformity measures. How does it affect the downstream task of uncertainty estimation? E.g. conformal prediction literature values smaller sets, is it useful for the UE task?
* In the literature on conformal prediction, a stronger property than marginal coverage (equation 1) is of interest: conditional coverage [1]. While this might be out of scope for the current work, was this property of the proposed method investigated?

**Limitations:**

I think that the majority of limitations are well discussed. However, I think that the current theoretical result being written in terms of the noise in score function is a significant limitation.

---

> ### Author Rebuttal · Authors · 2024-08-06
>
> We would like to thank the reviewer for engaging with our work. We tried to clarify two of the mentioned weaknesses/concerns, elaborate on one another, and address their questions.
>
> - **W1.** In our implementations, we ensure that each parameter of the Dirichlet distribution is at least one. We define the Dirichlet parameter as 1+NN(x), where NN(x) represents the non-negative output of the neural network. The intuition behind this is that, in the worst-case scenario where the output of the neural network is zero, we encounter a complete lack of knowledge, and this can be represented by setting the Dirichlet parameter vector to 1, which defines a uniform second-order distribution over all first-order distributions. We'll ensure to clarify this point in the paper.
> - **W2.** We stated the bounded noise assumption at the nonconformity score level to have a universal applicability for all nonconformity functions. For instance, when the Total Variation (TV) distance is used as the nonconformity function, the left-hand side of equation (11) in the paper can be rewritten as:
>     \begin{align*}
>      \mathbb{P}\left(|f(x, {\lambda}^{x})-f({x}, \tilde{\lambda}^{x} ) | < \epsilon \right) &=
>      \mathbb{P}\left(|d_{TV}(\hat{\lambda}^{x}, {\lambda}^{{x}})-d_{TV}(\hat{{\lambda}}^{{x}}, \tilde{{\lambda}}^{{x}} ) | < \epsilon \right) \geq \mathbb{P}\left(d_{TV}(\tilde{{\lambda}}^{{x}}, {\lambda}^{{x}}) < \epsilon \right)
>      \end{align*}
>      where the inequality holds according to reverse triangle inequality and $\hat{{\lambda}}^{{x}}$ denotes the prediction of the model. We can then use the Dvoretzky–Kiefer–Wolfowitz (DKW) inequality  to bound the total variation distance between the true categorical distribution and its empirical estimate from $m$ independent samples:
>     \begin{align*}
>     \mathbb{P}\left( d_{TV}(\tilde{{\lambda}}^{{x}}, {{\lambda}}^{{x}}) \leq \epsilon \right) \geq 1 - 2K \exp\left(-2m\epsilon^2\right).
>    \end{align*}
> Hence, when using TV as the nonconformity score for first-order approaches, the bounded noise in the nonconformity score reflects the noise arising from estimating the ground truth first-order distributions with only samples from that distribution. It is not clear how to provide such a bound for arbitrary nonconformity functions, which is why we presented the theoretical results under the assumption of bounded noise on the nonconformity level.
> - **W3.**  Please refer to our global response, in which we explained why we see such dependency and how we handled it.
> - **Q1.** As discussed in the response to W2, the DKW inequality can be used to bound the total variation distance between the true categorical distribution and its empirical estimate based on m independent samples, however, in the special case where m=1, this bound can be quite loose and, therefore, less useful. Interestingly, in Appendix E.2, we conduct experiments with synthetic data where we use the estimation of the ground-truth first-order distribution with varying values of m, including m=1. As illustrated in Figures 13 and 15, coverage is generally achieved in almost all cases. However, for lower values of m, this coverage comes with reduced efficiency, resulting in larger sets.
> - **Q2.** Please refer to the response provided for **W1**.
> - **Q3.** Please refer to our global response.
> - **Q4.** For the uncertainty interval figures, our goal was to present the information similarly to our credal sets, as shown in Figure 2, where we visualize the location of the ground truth distribution, the predicted distribution (either first-order or second-order, depending on the model), and the credal set. Analogously, we display the uncertainty interval together with the entropy of the ground truth distribution and the entropy of the predicted first-order distribution or the entropy of the mean of the Dirichlet distribution for the second-order predictor.
>     However, the entropy function is non-injective, meaning that two different distributions can have the same entropy. Therefore, the fact that the entropy of the ground truth is within the uncertainty interval does not necessarily imply that the credal set covers the distribution itself.
> - **Q5.** It is certainly desirable to achieve the coverage guarantee with the smallest possible set sizes. However, with a credal set, uncertainty depends on both the size and the location of the set. Broadly speaking, set size influences epistemic uncertainty, while location affects aleatoric uncertainty. The least uncertain scenario would be a singleton located at a corner of the simplex.
> - **Q6.** Investigating conditional coverage is definitely an interesting and important direction for future work in our paper. We anticipate this to be an even more challenging problem, as conditional coverage in standard conformal prediction (CP) pertains to the coverage given x or y (the label). In our case, however, it involves coverage given x or the first-order distribution.
> - **Limitations.** We hope our response to W2 has helped clarify the limitation.

---

> > ### Author Response · Authors · 2024-08-12
> >
> > Dear Reviewer LLVn,
> >
> > Given that we have addressed all your questions and the raised weaknesses, we are wondering if there are any remaining points that are unclear or require further explanation for us to clarify and improve our work.
> >
> > Best regards,
> >
> > The Authors

---

> > > ### Comment · Reviewer_LLVn · 2024-08-14
> > > **Rebuttal well received**
> > >
> > > Dear authors,
> > >
> > > your rebuttal was well received and you partially addressed my concerns. I will decide on the score changes after the discussion with other reviewers.

---

### Official Review · Reviewer_wGcP · 2024-07-11

**Soundness:** 4
**Presentation:** 3
**Contribution:** 4
**Rating:** 8
**Confidence:** 4

**Summary:**

The paper shows that if access to the (even noisy) estimates of the conditional categorical label distribution is provided, one can define a confidence region within the simplex of conditional probabilities, that is guaranteed (marginally) that the true aleatoric uncertainty (ground truth label distribution) is within the predicted credal set with $1 - \beta$ probability. The authors define various score functions based on distances in the probability space and likelihood.

**Strengths:**

The problem is really an interesting one. It is still interesting to have guaranteed uncertainty quantification and this paper is a step towards that goal.

The paper is technically sound. I did not detect flaws or issues in the paper at the end.

The experimental setup is complete.

**Weaknesses:**

Please see the questions.

**Questions:**

1. I believe that the results on the imprecise first-order data (theorem 3.2) can be enhanced leveraging the results from [1]. This study derives the conformal guarantee for the clean ground truth label, from the noisy prediction with a correction over the bias in noise (which in symmetric noise assumption it is 0.5). However, the result I mentioned concerns the prediction and adapting it to classification would be a different approach out of the context of this review.

2. Is there any way that the authors can derive a guarantee over the sampled labels as well? By that I mean deriving the conformal prediction based on labels sampled from the ground truth distribution (in cases that the non-soft label exists like CIFAR10) and an algorithm that returns prediction set from the credal sets.

3. If the answer to (Q2) is yes then what does the conformal quantile threshold look like in the simplex space?

4. Can we directly use some notion of uncertainty quantification (like log-likelihood) through the conformal risk control? I mean in Fig. 3. I see some of the credal sets are far from the ground truth distribution. Can the result enhance if you bound the TV distance in average via risk contol?

5. Are the scores that the authors used always producing symmetric convex halls? By symmetric I mean that the output is always a function of the predicted uncertainty and the computed quantile. If yes is there any way to remedy that?

[1] Guille-Escuret, Charles, and Eugene Ndiaye. "From Conformal Predictions to Confidence Regions." arXiv preprint arXiv:2405.18601 (2024).

**Limitations:**

I do not see any limitations that are not addressed or mentioned.

---

> ### Author Rebuttal · Authors · 2024-08-06
>
> Thank you for your kind words and feedback. We found your questions very interesting and did our best to elaborate on them.
> - **Q1.** Thank you for the reference; it is definitely an interesting and relevant work. To the best of our understanding, the main difference between the two papers is that [1] assumes additive noise on the level of labels while we are dealing with noise on the level of first-order distributions. It is not immediately clear to us how to incorporate the results from [1] into our context.
> - **Q2.** and **Q3.** Regarding an algorithm that generates prediction sets from credal sets, as noted in our response to W3 for reviewer gqZt, this topic has been explored in the field of imprecise probability. For instance, one approach involves excluding a label from the prediction set if its probability is dominated by other classes across all distributions within the credal set or the approach mentioned by reviewer gqZt [2]. However, extracting a conformal prediction set (with guarantees) from a conformal credal set is indeed an interesting follow-up research question stemming from our paper, and it is not an easy one to address immediately. More importantly, we believe that such an approach could greatly enrich the prediction sets by providing valuable insights into the uncertainty components (AU and EU).
> - **Q4.** The reason that some predictions do not cover the ground truth within the credal set is due to the specified error rate of Conformal Prediction (CP). While using conformal risk control will change the guarantee from coverage to risk, it's not immediately clear how and in what sense this can improve the results. However, it can definitely be considered an interesting future work.
> - **Q5.** This phenomenon occurs with credal sets constructed by the first-order predictor because, once the quantile is determined during the calibration step, all distributions within a certain distance are included in the set for any given point during test time. This is similar to the issue found in standard Conformal Prediction (CP) for regression when using absolute error as the nonconformity score, which is often criticized for its lack of adaptivity. In the CP for regression literature, one way to address this issue is through normalized nonconformity measures, where the score for each point is divided by a measure of its dispersion at that point [3]. In Appendix D of the paper, we introduced a normalized nonconformity score that divides the distance function by the entropy of the predicted distribution. Figure 10 demonstrates the successful improvement in the adaptivity of our proposed approach.
>
> [1] Guille-Escuret, Charles, and Eugene Ndiaye. "From Conformal Predictions to Confidence Regions."
>
> [2] M. Caprio et al., "Credal Bayesian Deep Learning."
>
> [3] H. Papadopoulos et al., "Normalized nonconformity measures for regression conformal prediction".

---

> > ### Comment · Reviewer_wGcP · 2024-08-13
> >
> > I thank the authors for their reply.
> >
> > After reading the authors response I believe that my questions are answered in a convincing way. Still, I will not change the score as it is already high. Here I break down my idea about the replies.
> >
> > 1. For Q1 I totally mentioned a new possible direction and therefore I do not expect the authors to modify the current state of the paper based on that. Their comment was clear and enough.
> >
> > 2. for Q2 and Q3: I believe the illustrative solutions work with a guarantee higher than 1 - alpha and not concentrated around it since the approach will be conservative. Intuitively I think that one simple approach is via sampling from the credal set and then defining the prediction set. But this problem is really confusing and addressing it is outside of the scope of this review/comment box.
> >
> > 3. I still think that one additional (and easy to experiment) study could be an ablation over different risk functions. But even without that I think the current state of the paper is acceptable.
> >
> > **Why not increase:** I think that the current score is already high enough and I do not think that the paper is an award-winning paper.
> >
> > **Why not decrease:** Despite the difference in my score to other reviewers, I still do not decrease my score; here I can address some key points from reviewers with lower scores and why I do not find their comments a reason to reject the paper:
> >
> > 1. **No comparison with other uncertainty quantification papers:** Other methods like deep-ensembles, Bayesian deep learning and dropout, provide uncertainty quantification without any notation of guarantee. Their output is a probability on the top predicted class and not a joint quantification of confidence over all classes. I think in general, comparing Bayesian UQ approaches with a frequentist method like conformal prediction is slightly off topic. Vanilla CP concerns ranking and therefore there is no priority among any two labels that are in the prediction set. Similarly, Conformal Credal Sets also should not prioritize any point inside the guaranteed credal set at first place so other UQ methods are not (at least trivially) comparable.
> >
> > 2. **Question on how to use credal sets:** It is pointed out by some reviewers that the credal sets (2nd order probability space) does not clearly show how to use in end-point prediction task. Although the method is a bit far from a good end-user experience, a new direction of work could be to interpret their results into meaningful probability alongside the top label etc.  Please note that the credal sets can be defined independent of the architecture of the model and training procedure. I believe leaving these interpretations aside, the question of how to use a frequentist approach to decide a probability over labels is still a strong question to answer.
> >
> > After all I also believe that some terminologies (like noisy labels) are a bit unclear and I strongly encourage the reviewers to rethink the writing and some terms used in the paper. This was also an encouragement (supported clearly by other reviews) to decrease the score since the paper does not deliver a clear intuition to someone who reads it without any background on basics of CP and its differences to non-frequentist approaches of uncertainty quantification.

---

> > > ### Author Response · Authors · 2024-08-13
> > >
> > > Thank you so much for engaging with our rebuttal. Your positive feedback is truly encouraging. We also greatly appreciate your thoughtful elaboration on points raised by other reviewers, which we found to be very helpful.

---

### Official Review · Reviewer_gqZt · 2024-07-11

**Soundness:** 2
**Presentation:** 2
**Contribution:** 2
**Rating:** 3
**Confidence:** 4

**Summary:**

The authors propose a method for predicting credal sets in the classification task, given training data labeled by probability distributions. The resulting conformal credal sets are guaranteed to be valid with high probability (without any assumptions on model or distribution). The applicability of the method on ambiguous classification tasks for uncertainty quantification is demonstrated.

**Strengths:**

1)	The methods is a novel and promising proposal.

2)	Experiments demonstrate interesting results of the proposal.

3)	The paper is overall well-structured, some parts are easy to follow.

**Weaknesses:**

1)	There are no baseline comparisons. For instance, seconde-oder distribution methods like Bayesian approaches and some limitations are mentioned  in the Introduction. It is  unclear what are the added gains of the proposed method in terms of uncertainty estimation, compared to the existing tools like Bayesian neural networks, deep ensembles or conformal prediction framework methods.

2)	Following point 1), Appendix F shows that the proposal can take up to a few hours to create the credal set at the test time compared to the training and calibration steps for a few minutes. From the practical point of view, it might be a significant drawback when the added gains are unclear.  In terms of the paper presentation, it would be better that the main body at least indicates that the computational cost is discussed in the Appendix.

3)	The paper seems not discuss how to predict the class index from the generated credal sets. Having a clear strategy of making class prediction would be a benefit for the end-users in practice. A recent paper [1] may provide some inspirations.

4)	Uncertainty quantification is a core element in this paper. However, from the results in Figure 4, 5, and 6, it is a bit hard conclude the quality of the uncertainty estimation. Why some down-stream tasks, like Out-Of-Distribution detection or Accuracy Rejection [2], are not used for the evaluation?

[1] Caprio, Michele, et al. "Credal Bayesian Deep Learning." arXiv e-prints (2023): arXiv-2302.
[2] Shaker, Mohammad Hossein, and Eyke Hüllermeier. "Ensemble-based uncertainty quantification: Bayesian versus credal inference." PROCEEDINGS 31. WORKSHOP COMPUTATIONAL INTELLIGENCE. Vol. 25. 2021.

**Questions:**

1)	In lines between 107-109: Could authors please explain more how the proposed method tackle the trade off achieving the high probability of courage (tend to increase the size of credal sets) and being informative?

2)	In lines between 124-127: Could authors please explain more why the Dirichlet distribution assumption is a meaningful learning? Would it also an biased estimation of uncertainty due to the lack of the ground-truth values?

4)	In lines 141-142: It is stated that both predictors can also be trained using zero-order data. Could the also add some baseline compares like deep ensembles, BNNs or conformal predictions, using zeor-order data to demonstrate the outperformance of the proposal?

5)	Could the authors indicate what is the potential application for the method?

**Limitations:**

Some limitations are discussed in the Limitations Section.
Additional limitations please refer to Weaknesses and Questions.

---

> ### Author Rebuttal · Authors · 2024-08-06
>
> We would like to thank the reviewer for their insightful comments and suggestions. We have made every effort to address and clarify their concerns in the following response.
>
> - **W1.** The comparison with other methods was not feasible primarily due to differences in data assumptions. While the methods mentioned in the review rely on zero-order data, our approach requires first-order data. Additionally, methods such as Bayesian Neural Networks (BNNs) or ensembles represent uncertainty using second-order probability distributions, whereas our approach utilizes credal sets. This fundamental difference in representation makes a direct comparison impossible. It's also worth noting that our approach is indeed a variant of conformal prediction specifically adapted to handle first-order data. The key advantage of our proposed conformalized credal sets is that they are designed to be valid in the sense that they cover the ground truth first-order distribution with high probability—a property that, to the best of our knowledge, is unique to our approach. Interestingly, when first-order calibration data is available, our approach can be applied to BNNs or ensembles to construct valid credal sets based on their predicted second-order probability distributions.
> - **W2.** We believe it's important to clarify the details in Appendix F. For CIFAR10-H, we discretized the simplex into 100,000 distributions. For each test data point in CIFAR10-H, creating credal sets—determining whether each of those 100,000 distributions belongs to the credal set across all four methods (TV, KL, WS, and Inner)—takes only 11 seconds. The majority of this time ($\sim$10.4 seconds) is attributed to the WS method, where we were unable to implement parallelization and had to iterate through all simplex distributions using a for loop. As a result, constructing all credal sets for the entire test set of 1,000 instances can take up to 3 hours. For this dataset, the calibration step and uncertainty quantification were completed in less than a second. We will ensure to clarify this in the paper to avoid any confusion.
> - **W3.** We fully agree with the reviewer that deriving a set of labels from a credal set is indeed a very interesting question, even though it is not the intention of our paper. This topic has been explored in various studies within the field of imprecise probability, such as the very interesting approach mentioned by the reviewer, or even simpler ones, such as the one where a label is excluded from the prediction set if its probability is dominated by other classes across all distributions within the credal set. In addition to these approaches, one could also apply the calibration step of the Monte Carlo Conformal Prediction method proposed by [1] in parallel, allowing for the construction of a prediction set alongside the credal sets.
> - **W4.** Please refer to our global response, specifically the **Evaluation of uncertainty quantification** paragraph.
> - **Q1.** This is indeed the desired property we're aiming for, and the same question applies to standard Conformal Prediction (CP). While CP naturally prevents prediction sets from becoming arbitrarily large by providing an upper bound on coverage probability under reasonable assumptions, the set size also depends on the chosen nonconformity score. In our work, we've explored and compared several scores, as shown in Figure 3 for the ChaosNLI dataset.
> - **Q2.** For second-order predictors, we need to assume a parametric distribution and estimate its parameters from the data. The Dirichlet distribution is a common choice, as it has been used in other second-order predictors like evidential deep networks [2].
> Regarding the second question, if the reviewer is referring to cases where the imprecise first-order distributions are biased estimates of the ground truth and we lack the bounded assumption of Theorem 3.2, our approach would be limited to the observations provided by these imprecise first-order distributions. Consequently, our uncertainty representation would be valid relative to these biased data, which ultimately reflect a bias toward the ground truth.
> - **Q3.** For the reasons why direct comparisons are not feasible, please refer to the response provided for W1. Note that learning models, such as first-order or second-order predictors, can be trained using zero-order data but not the credal predictor. The proposed credal predictor requires at least first-order data for calibration. For example, in our experiments with CIFAR-10H, we conformalize a probabilistic classifier trained on CIFAR-10 (zero-order) training data with CIFAR-10H (first-order) calibration data.
> - **Q4.** Our proposed method enables classifiers to represent uncertainty in a reliable manner through the use of a credal set. With this credal set, we can quantify both aleatoric and epistemic uncertainty, which can then inform decisions on the appropriate course of action. For instance, when epistemic uncertainty is high, it may be addressed by gathering more data. Conversely, if total uncertainty is primarily driven by high aleatoric uncertainty, additional data collection (e.g., consulting more experts) may not be beneficial and could be considered a waste of resources. We also demonstrated this with an illustrative experiment using synthetic data in Appendix E.1.
>
> [1] D. Stutz et al., "Conformal prediction under ambiguous ground truth."
>
> [2] M. Sensoy et al., "Evidential deep learning to quantify classification uncertainty."

---

> > ### Comment · Reviewer_gqZt · 2024-08-12
> > **Response to the rebuttals**
> >
> > Dear Authors,
> >
> > Thank you very much for your engagement in the rebuttal. After carefully reading your responses, I think my main concerns remain unresolved. For better clarity, I listed them as follows:
> >
> > **No baseline comparison with other existing uncertainty representation methods. The added gain of the proposal remains highly unclear.**
> >
> > As you indicated in Eq. (4) in the paper, we could train a standard neural network whose prediction is a single probability distribution from the first-order data with probabilistic ground truth by minimizing the cross entropy (Kullback-Leibler divergence). Therefore, deep ensembles can be built in this context. Similarly, I believe that Bayesian neural networks (BNNs) can also be trained using first-order data.
> >
> > Therefore, *'While the methods mentioned in the review rely on zero-order data, our approach requires first-order data'* in your response is not very convincing to me, because we could the existing baseline methods do not have limitations in using first-order data.
> >
> > As your proposed method requires at least first-order data for calibration, the comparison could be conducted in two aspects to enhance the contribution and soundness of your work potentially:
> >
> > a.	Train some existing second-order methods (e.g., deep ensembles, BNNs, etc.) on zero-order data and credal predictor on zero-order data with first-order data for calibration.
> >
> > b.	Train all the baselines on the first-order data.
> >
> > I acknowledge that credal sets are different representations of uncertainty compared to deep ensembles, BNNs, evidence-based deep learning, etc. However, some downstream tasks are widely used in the community to compare different uncertainty representations.
> >
> > For example, in the evidential deep learning paper [A], where Dirichlet distributions are used for uncertainty representation, they compared their methods with other representation methods, such as the BNN model and deep ensembles for performance comparison.
> >
> > As for credal representations, for instance, the paper [B] also compared credal inference with Bayesian approaches. The paper [C] also compared their method, a credal self-learning approach, with some strong baseline to convince the audience of the added gain of their proposal.
> >
> > [A]M. Sensoy et al., "Evidential deep learning to quantify classification uncertainty."
> >
> > [B] Shaker, Mohammad Hossein, and Eyke Hüllermeier. "Ensemble-based uncertainty quantification: Bayesian versus credal inference." PROCEEDINGS 31. WORKSHOP COMPUTATIONAL INTELLIGENCE. Vol. 25. 2021.
> >
> > [C] Lienen, Julian, and Eyke Hüllermeier. "Credal self-supervised learning." Advances in Neural Information Processing Systems 34 (2021): 14370-14382.
> >
> > *Therefore, I believe adding some baseline comparisons would strengthen the contribution of this work.*
> >
> > **Complexity discussion**
> >
> > Thanks for the clarity on the complexity discussion. 11 seconds per test point seems acceptable. I think by adding a cost reference (for instance a baseline), the audience (end-users) can better sense the added relative complexity overhead. *'For this dataset, the calibration step and uncertainty quantification were completed in less than a second.'*  Does it include the time cost of solving the optimization problem in Eq. (12)?
> >
> > **Deriving the class index (set) from the generated credal set as well as the potential applications**
> >
> > I thank the authors for elaborating on the possible ways of doing that. From a practical point of view, I believe it is necessary to have such a strategy. Compared to, for example, the Bayesian approach delivering a class index as the final prediction, conformal prediction generating class index sets, only have a credal set as the end can induce high complexity to derive a class prediction as well as interpret and apply the results to the downstream, e.g., decision problems. As the proposed method requires at least first-order data for calibration, could the author enlighten me on the potential application (e.g., the first-order data widely exists and is used for classification) of this method?
> >
> > I do think that the work is interesting and has potential. However, I think that it needs more work before publication.

---

> > > ### Author Response · Authors · 2024-08-12
> > >
> > > Thank you for engaging with our rebuttal. We're glad you find the work interesting and see its potential. We'd like to provide further details on the points you mentioned.
> > > - **Comparison to baselines**: We thought again about your remarks and would summarize our point of view as follows:
> > >     - Credal sets and second-order distributions are different uncertainty representations, and you also agree that a direct comparison is not possible -- it's like comparing apples with oranges. The same applies to uncertainty measures applied to these representations, which are of a different nature and not commensurable.
> > >     - Since a credal set cannot be turned into a second-order distribution, but a second-order distribution can be turned into a credal set in a more or less natural way via thresholding (like deriving a confidence interval from a density function), the only meaningful comparison we can envision is in terms of credal sets produced by different methods. What we could imagine, therefore, is a comparison of the credal sets produced by our method with the credal sets obtained, e.g., as the convex combination of the predictions of an ensemble of deep neural networks (trained on either zero- or first-order data), similar to what Shaker et al. did in reference [B] that you provided. The comparison could be done, e.g., in terms of the precision/efficiency and the coverage/validity of the credal sets, i.e., in a Pareto-sense. Before starting such experiments, it would be important to know whether that would be in your favor.
> > >     - Comparing the performance of different methods on downstream tasks is potentially also possible. However, we intentionally avoid such comparisons, which have their own problems and can compare uncertainty quantifications only indirectly -- in the literature, they are only used in the absence of first-order data (e.g., through measures such as coverage, efficiency, or other metrics presented in Figures 4 and 6 or the new measure proposed in our main rebuttal response). However, given access to first-order data, we can evaluate performance in a more direct way, which is clearly preferable. Also, shifting the focus to downstream performance at this point would change the scope of the paper substantially.
> > > - **Complexity discussion**: We're happy that our responses addressed your concern and appreciate your suggestion. Regarding your question, yes, finding the distributions with maximum and minimum entropy was included. Please note that we are using simplex discretization, which makes these calculations extremely fast and efficient.
> > >
> > > - **Deriving the class index (set)**: We're glad our responses were helpful. Regarding your question, as we mentioned in the paper, datasets in which multiple human experts annotate each data instance are becoming increasingly available in practice, including widely used benchmarks like ImageNet [1] and CIFAR-10 [2]. Crowdsourcing frameworks further facilitate such annotation mechanisms, making them even more accessible. (Please refer to our extended related work in Appendix A for additional references.)
> > > One particularly interesting and important case is in the medical domain, especially in dermatology [3], where multiple doctors evaluate the same image to classify a skin disease, often with conflicting opinions. This conflict can be aggregated into a probability distribution over labels, effectively representing aleatoric uncertainty, which is often overlooked.
> > >
> > > [1] L. Beyer et al., "Are we done with imagenet?"
> > >
> > > [2] J. C. Peterson et al., "Human uncertainty makes classification more robust"
> > >
> > > [3] D. Stutz et al., "Conformal prediction under ambiguous ground truth."

---

### Author Rebuttal · Authors · 2024-08-07

We want to express our gratitude to all four reviewers for their valuable feedback. We're encouraged that they found our paper important, interesting, and novel, and we appreciate their insightful comments. We would like to offer some general remarks, particularly concerning the evaluation part.

- **Uncertainty representation vs uncertainty quantification** We would like to emphasize that representing uncertainty through a credal set and decomposing it into aleatoric and epistemic components (i.e., uncertainty quantification) are two distinct tasks. This paper primarily focuses on proposing a novel method for representing uncertainty with a reliable credal set. To disentangle these uncertainties, we use a well-known measure despite some associated criticisms (see [1] for more details). The challenge of identifying an optimal measure for quantifying aleatoric and epistemic uncertainty within a credal set remains an active and important area of research.

- **Evaluation of uncertainty quantification (e.g., Figures 4 and 6)** Methods such as the accuracy-rejection curve and out-of-distribution (OOD) detection (mentioned by reviewer **gqZt**) are valuable for indirectly evaluating uncertainty quantification methods when only zero-order data is available. With first-order data, we can employ more effective methods to assess our approaches. Generally, with access to the ground truth first-order distribution, we should expect the following: When epistemic uncertainty (EU) is low, the quantified aleatoric uncertainty (AU) should be close to the entropy of the ground truth first-order distribution. However, when the EU is high, the quantified AU may vary, either close to or distant from this entropy. This is why we designed AU vs. EU plots, as shown in Figures 4 and 6.

  - The observed functional dependency (mentioned by reviewer **LLVn**) between quantified aleatoric and epistemic uncertainty is partly due to the measure we use. Generally, higher epistemic uncertainty implies a larger uncertainty interval, leading to a lower minimum value, i.e., aleatoric uncertainty. Our experiments with synthetic data in Appendix E.1 illustrate the behavior of the proposed credal sets and the performance of uncertainty quantification. As shown in Figure 12, this dependency disappears as epistemic uncertainty decreases.

  - Another unintended behavior, resulting from the lack of adaptivity in the first-order predictor, is evident in Figure 4, where low EU (below 0.5) is not paired with low AU. To observe such cases, very small credal sets concentrated around the corners would be needed. However, due to the lack of adaptivity, our method includes all distributions within a certain distance of the predicted one in the credal set, regardless of its position. This issue is similar to the problem in standard Conformal Prediction (CP) for regression when using absolute error as the nonconformity score, which is often criticized for its lack of adaptivity. In the CP for regression literature, normalized nonconformity measures address this issue by dividing the score for each point by a measure of its dispersion [2]. In Appendix D of the paper, we introduced a normalized nonconformity score that divides the distance function by the entropy of the predicted distribution. Figure 10 demonstrates the successful improvement in the adaptivity of our proposed approach.

- **New figures** If the current figures are still not as clear as we’d like, we are happy to enhance the readability by including additional figures similar to those attached to this rebuttal. We can plot the absolute difference between the quantified aleatoric uncertainty and the entropy of the ground truth distribution against the quantified epistemic uncertainty. This approach might make it easier to see that we expect a low error in this difference when epistemic uncertainty (EU) is low and a broader range of error when EU is high. We’re flexible and open to making such improvements to ensure the results are as clear and informative as possible.

[1] E. Hüllermeier et al., "Quantification of credal uncertainty in machine learning: A critical analysis and empirical comparison"

[2] H. Papadopoulos et al., "Normalized nonconformity measures for regression conformal prediction"

---

### Decision · Program_Chairs · 2024-09-25

**Decision:**

Accept (poster)

**Comment:**

The paper presents a new method for predicting credal sets in classification tasks, aiming to represent both aleatoric and epistemic uncertainty with high probability guarantees. The method is grounded in solid theory and is validated through experiments on datasets like ChaosNLI and CIFAR10-H. However, the paper received mixed feedback from reviewers. Some praised its novelty and effective uncertainty representation, while others pointed out the lack of comparisons with other uncertainty quantification methods and raised concerns about practical applicability, computational complexity, and the key bounded noise assumption.

In their rebuttal, the authors tried to addressed these concerns by explaining their theoretical contributions and the difficulties of making direct comparisons with other methods. They acknowledged the paper's limitations but defended their approach, emphasizing its unique aspects.

Overall, although the paper would benefit from further empirical comparisons,  the paper introduces a truly novel approach for uncertainty quantification, which is worth to be explored by the community.